# Spontaneous redox continuum reveals sequestered technetium clusters and retarded mineral transformation of iron

Daria Boglaienko [1], Jennifer A. Soltis [1], Ravi K. Kukkadapu [1], Yingge Du [1], Lucas E. Sweet[1], Vanessa E. Holfeltz [1], Gabriel B. Hall [1], Edgar C. Buck [1], Carlo U. Segre [2], Hilary P. Emerson [1], Yelena Katsenovich[3] & Tatiana G. Levitskaia [1✉]

The sequestration of metal ions into the crystal structure of minerals is common in nature. To date, the incorporation of technetium(IV) into iron minerals has been studied predominantly for systems under carefully controlled anaerobic conditions. Mechanisms of the transformation of iron phases leading to incorporation of technetium(IV) under aerobic conditions remain poorly understood. Here we investigate granular metallic iron for reductive sequestration of technetium(VII) at elevated concentrations under ambient conditions. We report the retarded transformation of ferrihydrite to magnetite in the presence of technetium. We observe that quantitative reduction of pertechnetate with a fraction of technetium(IV) structurally incorporated into non-stoichiometric magnetite benefits from concomitant zero valent iron oxidative transformation. An in-depth profile of iron oxide reveals clusters of the incorporated technetium(IV), which account for 32% of the total retained technetium estimated via X-ray absorption and X-ray photoelectron spectroscopies. This corresponds to 1.86 wt.% technetium in magnetite, providing the experimental evidence to theoretical postulations on thermodynamically stable technetium(IV) being incorporated into magnetite under spontaneous aerobic redox conditions.

[1] Pacific Northwest National Laboratory, Richland, WA 99352, USA. [2] Illinois Institute of Technology, Chicago, IL 60616, USA. [3] Florida International University, Miami, FL 33174, USA. ✉email: Tatiana.Levitskaia@pnnl.gov

Natural minerals can host, or incorporate into their crystal structure, different elements and function as sinks attenuating environmental transport of heavy metals and radionuclides. Garnet, gypsum, calcite, muscovite, rutile and other mineral phases are found to contain incorporated Tc, Ru, Cs, U, Np, and Pu[1]. Iron minerals, such as iron oxides (magnetite, maghemite, and hematite), and iron oxyhydroxides (goethite, lepidocrocite, ferrihydrite, etc.) also incorporate various metal cations, in particular, substitution of $Fe^{3+}$ by $Al^{3+}$ can reach up to 33 mol.% in natural weathered goethite and up to 15 mol.% in maghemite in tropical soils[2]. Computational studies on $Tc^{4+}$ incorporation in magnetite revealed thermodynamic feasibility to obtain up to 5 wt.% of incorporated $Tc^{4+}$ with higher stability around 1–3 wt.%[3], however, no experimental studies were carried out to support the upper limit of Tc incorporation in the ambient uncontrolled environment. Further, formation of iron minerals is complex as it yields an array of transformation pathways for different stable oxides/oxyhydroxides[2]. This allows for design of plentiful systems targeting a specific mineral formation in which an element of interest is being incorporated. Particularly, there is practical significance to employ widely available metallic iron ($Fe^0$), encouraging investigation of its oxidative transformation with the possibility of concomitant sequestration of radionuclides undergoing simultaneous reduction, e.g., Tc, in a system under ambient conditions where only a limited number of controlled parameters are imposed to resemble natural pathways.

The radioisotope of technetium, $^{99}Tc$, is a $\beta^-$ emitter of predominantly anthropogenic origin that generates environmental risk in sites impacted by cold war era development of nuclear weapons (e.g., Hanford site WA, USA;[4] Mayak, Russia[5]) or nuclear accidents (e.g., Chernobyl, Ukraine[6]), and its environmental impact is amplified by a long half-life (213,000 years) and redox-dependent mobility with high solubility of the pertechnetate anion ($TcO_4^-$) in aerobic conditions[7]. Under anoxic conditions Tc reductively precipitates as hydrous $TcO_2$ and can be retained in iron-rich sediments even upon consequent exposure to air[8]. The possibility of utilizing iron-based phases as reductive separation and sequestration agents initiated numerous studies on Tc immobilization into iron minerals[9–19]. The majority of these studies investigated reductive removal of Tc from solution via structural incorporation into in situ formed mineral iron oxide/oxyhydroxite phases, i.e., ferrihydrite, magnetite, hematite, and goethite, starting with soluble homogeneous $Fe^{2+}$ precursors[20]. Immobilization of Tc within the crystalline structure of a host helps to achieve recalcitrant oxidative leaching of $Tc^{4+}$, and synthesized $Tc^{4+}$-doped magnetite, hematite and goethite that have been evaluated as durable waste forms for long-term geological disposal demonstrated relatively low Tc release rates[13,14,21]. When structurally incorporated into in situ formed magnetite, $Tc^{4+}$ was prone to remain in the reduced state even upon oxidation of magnetite[15]. $Tc^{4+}$-doped magnetite ($Tc_{0.06}Fe_{2.94}O_4$) can be synthesized with 2.5 wt.% Tc by dissolving a small amount of iron powder in a denitrated solution of Tc[21], however, incorporation of $Tc^{4+}$ into pre-synthesized magnetite was found to be sensitive to its initial concentration[16]. Although in situ formed magnetite was more efficient for Tc reductive sequestration than the pre-synthesized one, these studies have been conducted in anoxic conditions. Iron oxidation in anoxic vs. oxic conditions would take a different route with water or dissolved oxygen serving as an oxidant, leading to dissimilar iron oxidation pathways[2].

There are limitations of utility of the homogeneous in situ synthesis of iron minerals for the practical separation systems design when it utilizes prone-to-oxidation $Fe^{2+}$, as it must be carried out in a controlled pH and redox sensitive environment, and therefore can't be accomplished in complex aqueous matrices

directly, necessitating prior separation of Tc from the original host streams. A valuable alternative is in situ mineral transformation of a metallic iron, $Fe^0$, or zero valent iron (ZVI), potentially offering continuum reductive separation and structural incorporation of Tc under ambient conditions from complex aqueous electrolyte mixtures over wide pH and concentration ranges, albeit via poorly understood mechanisms. Overall, ZVI is a readily available strong reductant that has been proposed for decontamination and remediation strategies, including reductive removal of radioactive contaminants, uranyl and pertechnetate, in groundwater[22–24]. ZVI (iron powder, nano-iron, and steel coupons) exhibits effective reductive separation of Tc[25–27], and is one of the materials compatible with immobilization and stabilization of the separated Tc for long-term disposal, but its application can be modulated by Tc anticorrosive properties[28,29], and has not been investigated for high Tc loading. Different iron oxides/ oxyhydroxides can be formed during reaction of metallic iron in aerobic conditions, and their transformation pathways are influenced by a variety of factors, including presence of anions[2], where effect of $TcO_4^-$ at moderately high concentration has not been studied yet. In addition, ZVI materials have shown different redox kinetics, depending on particle size and related to their manufacturing method[27]. Thus, the behavior of an iron system requires additional investigation for each unique setting and contaminant of concern.

Here, we seek a fundamental understanding of the in situ formation of iron oxide phases in the aerobic environment governed by the redox equilibria processes. We study a heterogeneous oxic system containing solid $Fe^0$ and aqueous $TcO_4^-$ initiating surface redox reactions associated with $Fe^0$ dissolution, followed by in situ co-precipitation with $Tc^{4+}$ reduction products. Our previous testing of a wide range of commercial iron products[27] shows that granular ZVI exhibits 99% Tc-removal efficiency indicating kinetics suitable for time-dependent structural studies on formation and transformation of iron minerals in the presence of Tc. The NaCl solution matrix is chosen in order to supply an electrolyte that promotes iron oxidation[2] and avoids complexation with $TcO_4^-$, i.e., without interfering with the $Tc^{7+}$ reduction process[30]. All experiments of reductive sequestration of Tc with granular ZVI and comprehensive characterization of solid phases are conducted at Fe:Tc molar ratio 53:1, or 3.3 wt.% Tc. The results show structural incorporation of $Tc^{4+}$ at the high loading and reveal effect of Tc on transformation of iron mineral phases.

## Results

**Reductive removal of pertechnetate**. After 25 days of contact time of granular ZVI with 80 mM NaCl solution containing 17 mM $TcO_4^-$, 99.8% of the aqueous Tc was removed from the solution (aqueous fraction of Tc 0.0010 ± 0.0004). In the presence of Tc, ZVI granules exhibited distinct time-dependent changes observable by color (Supplementary Fig. 1a, b), transitioning from brown-yellowish after a week to smaller black granules after a month from the start of the experiment. In the absence of Tc, the sample of ZVI reacted with 80 mM NaCl solution turned black within one week after the start of the experiment (Supplementary Fig. 1c). Both samples with and without Tc had a fraction of newly formed black suspended fines (likely magnetite), separated during several runs of consecutive centrifugation, and analyzed as a bulk solid phase with the rest of solids per each sample. The concentration of Tc in the supernatant was 0.01 mM of $TcO_4^-$. After centrifugation, samples were washed with deionized water, and the washing solution had slightly higher Tc concentration (0.08 mM) compared to the supernatant. This suggested possible re-oxidation or resuspension of $Tc^{4+}$.

**Table 1 Linear combination fitting.**

| | XANES weight fraction (one SD error on the last significant figure) | XANES Chi$_r^2$ | EXAFS weight fraction (one SD on the last significant figure) | EXAFS Chi$_r^2$ |
|---|---|---|---|---|
| ZVI-1-A | | | | |
| $TcO_4^-$ | 0.11 (1) | | 0.10 (1) | |
| $TcO_2 \bullet nH_2O$ | 0.55 (6) | 0.009 | 0.48 (3) | 0.04 |
| $Tc^{4+}$ in $Fe_3O_4$ | 0.34 (6) | | 0.42 (4) | |
| ZVI-1-B | | | | |
| $TcO_4^-$ | 0.21 (2) | | 0.15 (2) | |
| $TcO_2 \bullet nH_2O$ | 0.45 (9) | 0.021 | 0.45 (4) | 0.05 |
| $Tc^{4+}$ in $Fe_3O_4$ | 0.34 (9) | | 0.40 (4) | |
| ZVI-2-A | | | | |
| $TcO_4^-$ | 0.13 (1) | | 0.10 (1) | |
| $TcO_2 \bullet nH_2O$ | 0.55 (6) | 0.010 | 0.52 (3) | 0.03 |
| $Tc^{4+}$ in $Fe_3O_4$ | 0.32 (6) | | 0.39 (3) | |
| ZVI-2-B | | | | |
| $TcO_4^-$ | 0.20 (2) | | 0.15 (1) | |
| $TcO_2 \bullet nH_2O$ | 0.46 (9) | 0.020 | 0.47 (3) | 0.05 |
| $Tc^{4+}$ in $Fe_3O_4$ | 0.34 (9) | | 0.38 (4) | |

As investigated previously, ZVI materials manufactured by different methods exhibit different kinetics of Tc removal, attributed to variable surface properties and rates of $Fe^0$ oxidation and dissolution[27]. Among the wide variety of ZVI materials tested previously, granular iron belonged to a group with superior $Tc^{7+}$ reduction efficiency, and it exhibited the least propensity for $Tc^{4+}$ re-oxidation within this group. For this material, the profile of $Fe^0$ solubilization in 80 mM NaCl solution and the corresponding pH and ORP time dependencies had been studied earlier[27]. Here, the pH value of 10.3 measured 25 days after the start of experiment indicated the prevalence of iron dissolution reactions[2].

X-ray absorption near edge structure (XANES) analysis from different locations (A and B) on two iron granules (ZVI-1 and ZVI-2) showed that both $Tc^{4+}$ and $Tc^{7+}$ were present in the solid phase after one month of contact with ZVI (Table 1); however, partial oxidation of $Tc^{4+}$ during sample preparation, storage, and analysis cannot be ruled out. Linear combination fitting (LCF) was performed for XANES (21 to 21.4 keV) and extended X-ray absorption fine structure (EXAFS) spectra (3 < k < 12), and LCF combinatorics showed better statistical results for three components, with R-factor and Chi$_r^2$ almost two times smaller for three components vs. two components (Supplementary Table 1). Moreover, principal component analysis (PCA) was supportive of three components (Supplementary Fig. 2); hence, three standards were kept for LCF (Fig. 1a; Table 1), two of which were $Tc^{4+}$ with different coordination environments ($Tc^{4+}$ hydrated oxide and $Tc^{4+}$ in magnetite). These results revealed that approximately 50% of Tc was present as $TcO_2 \bullet nH_2O$ in addition to 34% of Tc in $Tc^{4+}$ phases associated with iron oxides. The remaining small fraction of Tc was associated with adsorbed $Tc^{7+}$ phases with a larger fraction present in B locations of both granules, where more Tc was associated with the solid phase, as seen from the false-color abundance maps, Fig. 1b (11–13% in A locations vs. 20–21% in B locations, Table 1).

We further performed high-resolution X-ray photoelectron spectroscopy (XPS) analysis of the ZVI granules (Fig. 2). Peak fitting of the Tc 3d spectrum required three pairs of doublets, whose lower binding energies ($3d_{5/2}$) were determined to be 256.4 eV, 257.4 eV, and 259.6 eV. While the peaks at 256.4 eV and 259.6 eV were assigned to $Tc^{4+}$ (35.3%) and $Tc^{7+}$ (32.8%), respectively[31,32], the peak at 257.4 eV suggested $Tc^{4+}$ in a different local environment (~31.9%) that is supported by XANES analysis (Fig. 1a and Table 1).

**Structural characterization of solid phase.** Powder X-ray diffraction (PXRD) analysis of ZVI granules contacted with 80 mM NaCl solution with and without $TcO_4^-$ was indicative of magnetite in all samples, with a more prominent magnetite pattern in samples without Tc, Fig. 3ab (data with Rietveld refinement are in Supplementary Fig. 3). Here, PXRD patterns did not allow for the discrimination between magnetite and maghemite, hence, maghemite formation was not excluded. In addition, the single crystal diffractogram of the one-month contacted ZVI sample without Tc revealed an additional iron phase, ferrihydrite (Supplementary Fig. 4); the sample with Tc did not produce a good signal.

XPS core level Fe 2p spectrum (Supplementary Fig. 5) showed a profile characteristic of $Fe^{3+}$ [33,34], with an Fe $2p_{3/2}$ peak at 711.2 eV and a signature satellite peak at ~719 eV, which may belong to maghemite. It should be noted that XPS is a surface sensitive technique with a sampling depth of <10 nm, and oxidized surface layers, i.e., magnetite oxidized to maghemite, may not be representative of the bulk.

Mössbauer spectroscopy confirmed the presence of non-stoichiometric, or partially oxidized, magnetite. Magnetite ($Fe_3O_4$) is an inverse spinel which has stoichiometry of 8 tetrahedral $Fe^{3+}$, 8 octahedral $Fe^{3+}$, 8 octahedral $Fe^{2+}$, and 32 O atoms with $Fe^{2+}/Fe^{3+}$ ratio equal to 0.5. Oxidation of magnetite leads to formation of maghemite ($\gamma$-$Fe_2O_3$) with a unit cell of 8 tetrahedral $Fe^{3+}$, 13 1/3 octahedral $Fe^{3+}$, 32 O atoms, and 2 1/3 vacancies for charge balance at octahedral sites, and its stoichiometric $Fe^{2+}/Fe^{3+}$ ratio is 0 at complete oxidation of magnetite. The experimental $Fe^{2+}/Fe^{3+}$ stoichiometry can be determined using Mössbauer signals as $Fe^{2+}/Fe^{3+} = 0.5$ $^{oct}Fe^{2.5+}/$ ($0.5$ $^{oct}Fe^{2.5+}$ + $^{oct,tet}Fe^{3+}$), where $^{oct,tet}Fe^{3+}$ is an overlapped sextet signal from $^{tet}Fe^{3+}$ and $^{oct}Fe^{3+}$ in partially oxidized samples, and $^{oct}Fe^{2.5+}$ is a sextet of octahedrally coordinated $Fe^{2+}$ and $Fe^{3+}$ [35].

Two sextets of octahedral-tetrahedral $Fe^{3+}$ and octahedral $Fe^{2.5+}$ were well-distinguished in the room temperature Mössbauer spectra of ZVI samples not exposed to Tc (sample oxidized for one week: Supplementary Fig. 6a; and sample oxidized for one month: Fig. 4a). Longer contact time led to partial oxidation of magnetite ($Fe^{2+}/Fe^{3+} < 0.50$), as the $Fe^{2+}/Fe^{3+}$ ratio decreased from 0.47 (one week contact time) to 0.39 (one month contact time), suggesting an effect of the topotactic transformation of magnetite, i.e., maghematization process[2]. A time-dependent

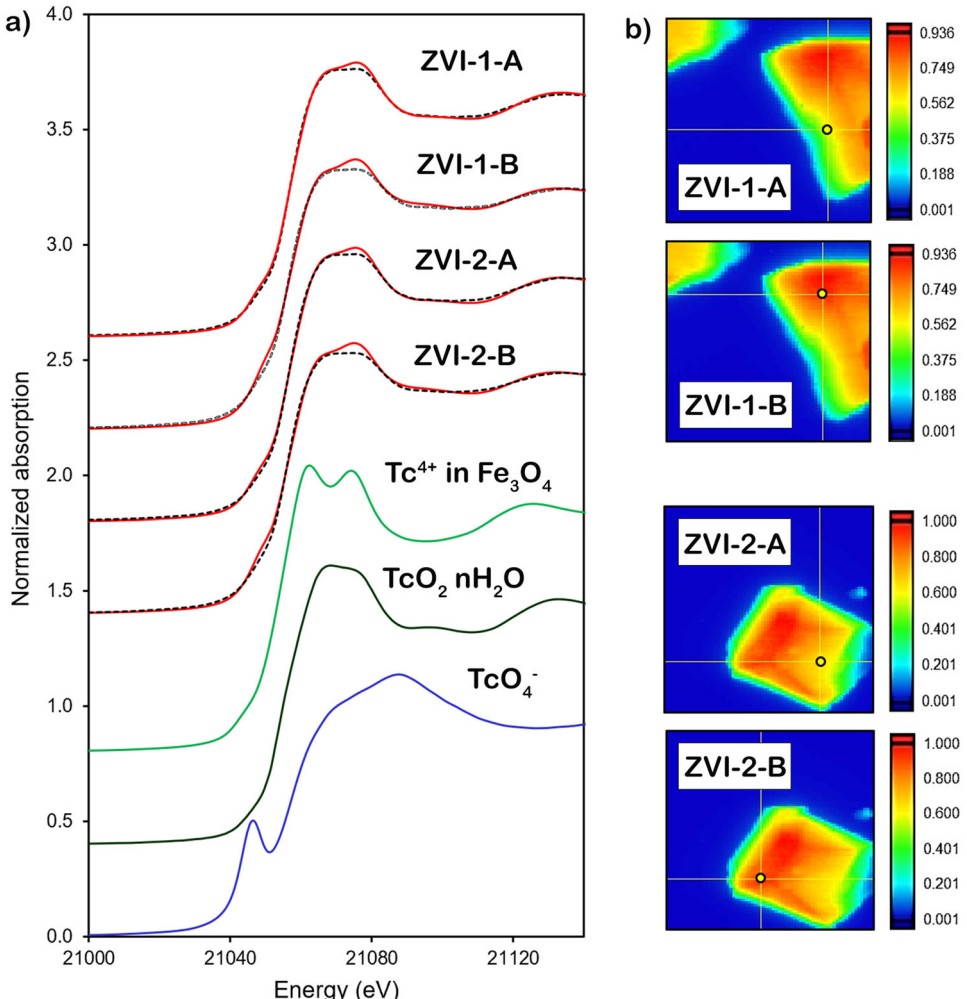

**Fig. 1 X-ray absorption near edge structure analysis. a** Tc K-edge XANES spectra for granular ZVI reacted with 17 mM $TcO_4^-$ in 80 mM NaCl for one month. Dotted black lines are data, solid red lines are fit; $Tc^{4+}$ in $Fe_3O_4$ (magnetite), $Tc^{4+}$ as $TcO_2 \bullet nH_2O$, and $Tc^{7+}$ as $TcO_4^-$ are standards. **b** X-ray fluorescence false-color images showing XAFS scans' spatial locations (indicated by the circle) and Tc abundance on ZVI particles. XAFS data were collected on two ZVI particles, 1 and 2, in two locations, A and B. The color scales are arbitrary intensities of fluorescent X-rays with the highest intensity assigned red color and number 1.0, and the lowest intensity— dark blue color and number 0.001.

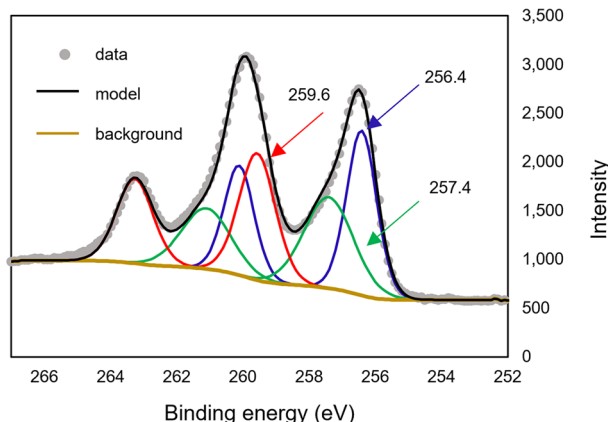

**Fig. 2 X-ray photoelectron spectroscopy analysis.** Scan of granular ZVI contacted with 17 mM $TcO_4^-$ in 80 mM NaCl for one month (Tc 3d spectrum).

increase of the magnetite-maghemite content in the samples without Tc was accompanied by the expected decrease of the $Fe^0$ phase (Supplementary Table 2).

Room temperature Mössbauer spectra of the ZVI samples contacted with Tc for one week and one month were both dominated by $Fe^{3+}$ doublets at 0–0.78 mm/s (Supplementary Fig. 6b) and at 0–0.72 mm/s (Supplementary Fig. 6c), respectively, that could be attributed to a mixture of phases, such as nano-sized magnetite[36], nano-sized maghemite[37], lepidocrocite[38], nano-sized goethite[39], or ferrihydrite[10]. The PXRD analysis excluded the dominant presence of goethite and lepidocrocite (Fig. 3); ferrihydrite was not easily evident due to its poorly crystalline nature, but its presence was supported by the results from single crystal analysis for the ZVI sample without Tc (Supplementary Fig. 4). Liquid nitrogen Mössbauer spectroscopy measurements allow identification of nano-sized magnetite and poorly crystallized maghemite resolved as a sextet due to superparamagnetic relaxation[36,37]. For this purpose, additional spectra of the one-month aged sample were taken at 77 K and provided evidence of a $Fe^{3+}$ doublet at 0–0.81 mm/s, comprising 27% of total iron and proving presence of ferrihydrite; the remaining spectral features were resolved as the sextets of octahedral-tetrahedral $Fe^{3+}$ and octahedral $Fe^{2+}$ (Fig. 4b). Decrease of the $Fe^{3+}$ doublet area and

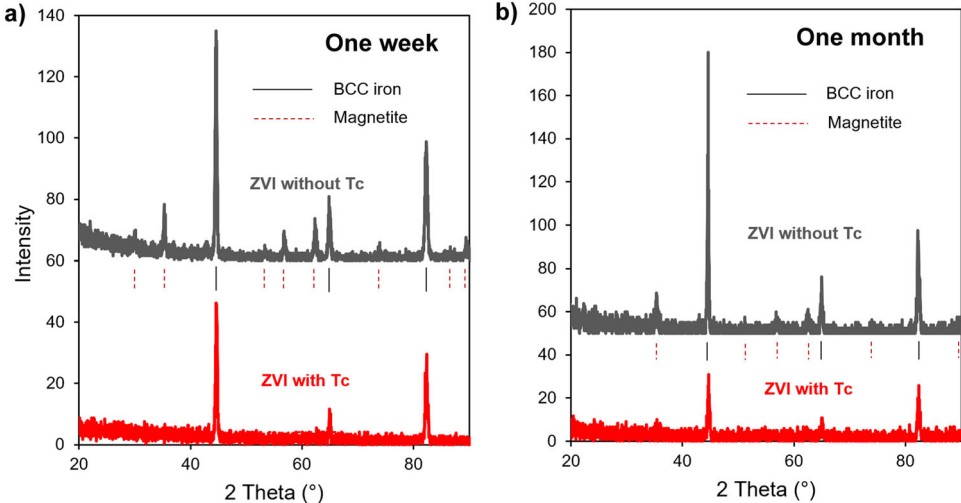

**Fig. 3 Powder X-ray diffraction patterns.** Granular ZVI reacted with and without 17 mM $TcO_4^-$ in 80 mM NaCl for a different period of time: **a** one week contact time; **b** one month contact time. Peak indication lines for non-reacted ZVI (BCC iron): solid black line, and for magnetite: dashed red line. Rietveld refinement results are given in Supplementary Fig. 3.

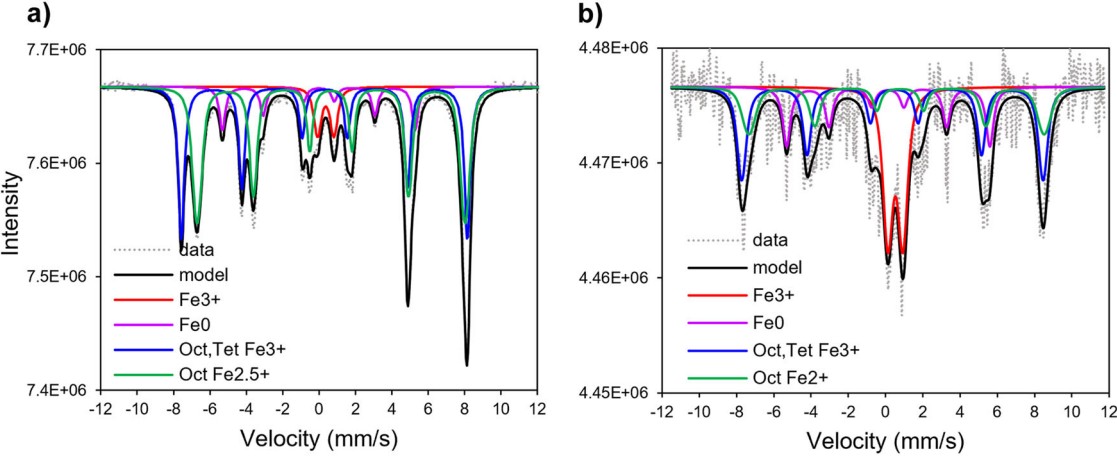

**Fig. 4 Mössbauer spectra.** Granular ZVI reacted with and without 17 mM $TcO_4^-$ in 80 mM NaCl for one month: **a** 298 K spectrum of ZVI reacted without Tc; **b** 77 K spectrum of ZVI reacted with Tc.

increase of the $Fe^{3+}$ and $Fe^{2+}$ sextet areas relative to the room temperature measurements is attributed to the presence of nano-sized magnetite and/or maghemite. Another interesting observation is that the $Fe^{2+}/Fe^{3+}$ ratio in the room temperature spectrum was 0.33 compared to 0.39 for the analogous sample without Tc (Supplementary Table 2), which signifies a higher fraction of octahedral $Fe^{2+}$ oxidized to $Fe^{3+}$, hence, higher content of non-stoichiometric magnetite or magnetite-maghemite mixture in the sample with Tc. Distinction between non-stoichiometric magnetite and magnetite-maghemite mixture is known to be problematic[40], and we further refer to it here as non-stoichiometric magnetite. The maghematization process is accompanied by the formation of vacancies during oxidation and loss of $Fe^{2+}$. This process is energetically favorable for $Tc^{4+}$ incorporation, where $Tc^{4+}$ substitutes $Fe^{2+}$ in the octahedral sites, and loss of two $Fe^{2+}$ atoms satisfies the charge balance[3]. Moreover, as shown by Marshall et al.[15], $Tc^{4+}$ incorporated into magnetite is recalcitrant to oxidation and release during long-term transformation of magnetite to maghemite.

Furthermore, comparison of the Mössbauer spectra corresponding to the ZVI samples exposed to NaCl solution with and without Tc for one month revealed higher fractions

of the unreacted $Fe^0$ and $Fe^{3+}$ (ferrihydrite) in presence of Tc (Supplementary Table 2; Supplementary Fig. 6). It should be noted that this analysis was primarily done for the purpose of qualitative assessment of the oxidized iron mineral phases.

**Microscopy analysis**. Cubo-octahedral morphology pertinent to magnetite[2,41] was observed by scanning electron microscopy (SEM) in the sample contacted with Tc for one month (Fig. 5). The surface of the iron oxide, formed as a result of ZVI dissolution and reprecipitation, was uniformly covered with Tc, as seen via energy dispersive X-ray spectroscopy (EDS) on particles contacted with Tc for one week (Supplementary Fig. 7) and on particles contacted with Tc for one month (Fig. 5). Moreover, Tc association with iron oxide (i.e., non-stoichiometric magnetite) was both on the surface and within iron oxide particles. Analysis of the lamella prepared by focused ion beam (FIB) showed that Tc was incorporated and homogeneously distributed well below the particle surface (Fig. 6). However, EDS performed at the nano scale revealed heterogeneous areas with local enrichment of Tc (Fig. 7).

High resolution scanning transmission electron microscopy (HR-STEM) analysis identified regions with the crystal structure consistent with maghemite or magnetite which cannot be distinguished because of close structural similarities (Fig. 7c). According to the Mössbauer spectroscopy results, the material in the images could be non-stoichiometric magnetite or a magnetite-maghemite mixture. In addition, EDS analysis of the edge of a particle revealed an area (approximately $100 \times 200$ nm) with little to no Tc present (Supplementary Fig. 8) which was also confirmed via HR-STEM to be either magnetite or maghemite.

Further, HR-STEM analysis of the ZVI sample contacted for one month without Tc was consistent with the presence of magnetite or maghemite (Supplementary Fig. 9).

The presence of other iron mineral phases was not detected with STEM analysis, however the lamella did not represent the bulk sample (Supplementary Fig. 10). The amorphous powder-like solids were not suitable for FIB extraction, thus, additional iron mineral phases cannot be excluded. Scanning electron micrographs of ZVI samples with and without Tc showed larger particles in the sample without Tc, and more of the powder-like iron oxidation product (i.e., ferrihydrite) in the sample with Tc (Supplementary Fig. 11). Possibly, retarded transformation of ferrihydrite to magnetite resulted in smaller particles in the sample with Tc.

**Extended X-ray absorption fine structure analysis.** Extended X-ray absorption fine structure (EXAFS) data analysis and modeling were performed to clarify $Tc^{4+}$ association with iron oxides (Fig. 8, Table 2). Even though non-stoichiometric magnetite or magnetite and maghemite mixture is referred to here, due to their structural similarities, a modified crystal structure of magnetite was used for the fitting model. The octahedral atom of iron (Fe1) was set as a core atom substituted by Tc. The coordination number (CN) of both octahedral Fe1 and tetrahedral Fe2 was fixed at 6.0 in accord with the crystal structure of magnetite. Theoretically, the first shell of octahedrally coordinated $Tc^{4+}$ is represented by six O atoms in both oxide and magnetite. The majority of the EXAFS studies on Tc incorporation into magnetite fixed CN to 6.0[15,16,30,42], however, our data did not result in a good fit with 6O atoms, possibly due to a fraction of the Tc being on the surface of the iron oxide nanoparticles where its coordination numbers can be reduced. While the CN for ZVI-1-A and ZVI-2-A (4.4 and 4.1, respectively) are reasonable, accounting for the mixture of $Tc^{7+}$ and $Tc^{4+}$ and surface fraction of $Tc^{4+}$, the CN for ZVI-1-B and ZVI-2-B are quite low (3.1 and 2.7, respectively), but the data for these scan locations correspond to the elevated amounts of Tc (false-color images with Tc abundance on ZVI granules, Fig. 1b),

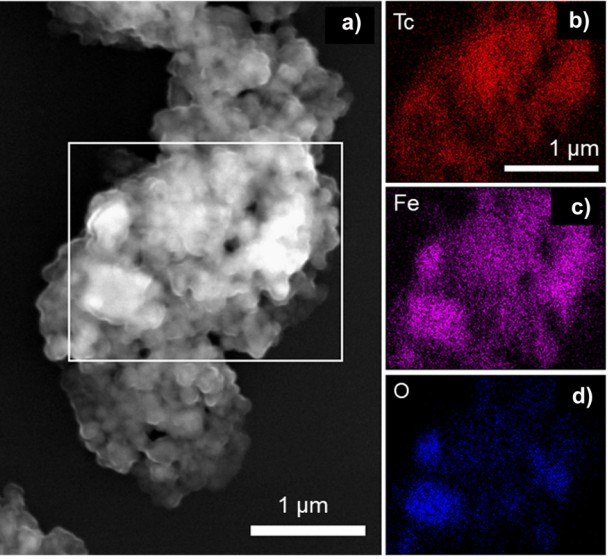

**Fig. 5 Microscopy analysis.** Granular ZVI reacted with 17 mM $TcO_4^-$ in 80 mM NaCl for one month. **a** SEM micrograph of the iron oxide as a result of ZVI granules dissolution/reprecipitation; **b–d** EDS maps of Tc (**b**), Fe (**c**), and O (**d**). The maps were collected from the area outlined in white on the SEM image (**a**).

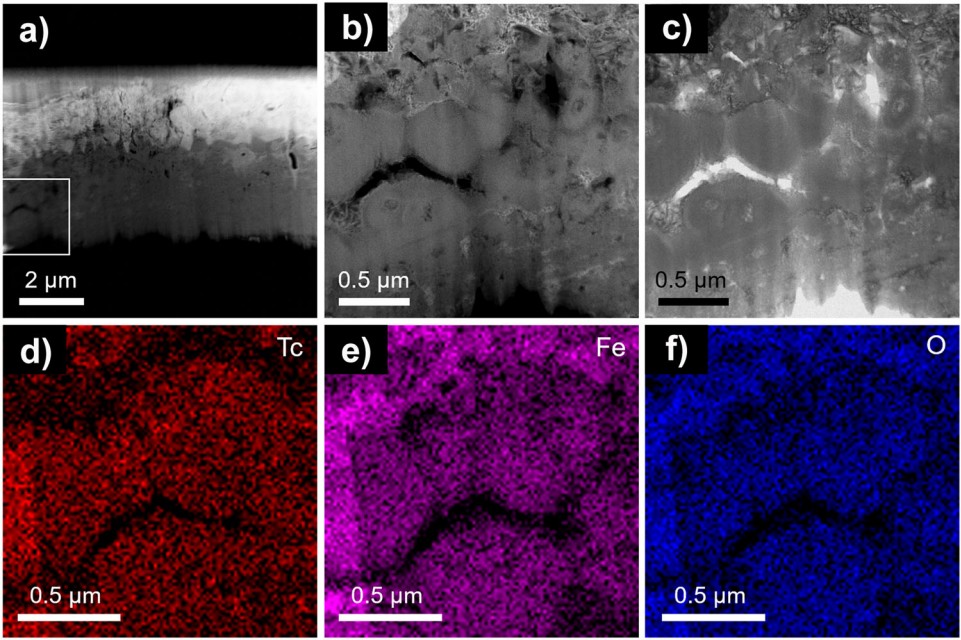

**Fig. 6 Microscopy analysis of the lamella at micron scale.** Granular ZVI reacted with 17 mM $TcO_4^-$ in 80 mM NaCl for one month. **a** HAADF STEM micrograph of Tc-containing lamella; **b** higher magnification HAADF image of the area outlined in white on the image (**a**); **c** bright field STEM image collected concurrently with (**b**); **d–f** EDS maps of the area shown in **b** and **c** indicate homogeneous distribution of Tc within oxidized ZVI. The mineral phases shown in this area could not be identified.

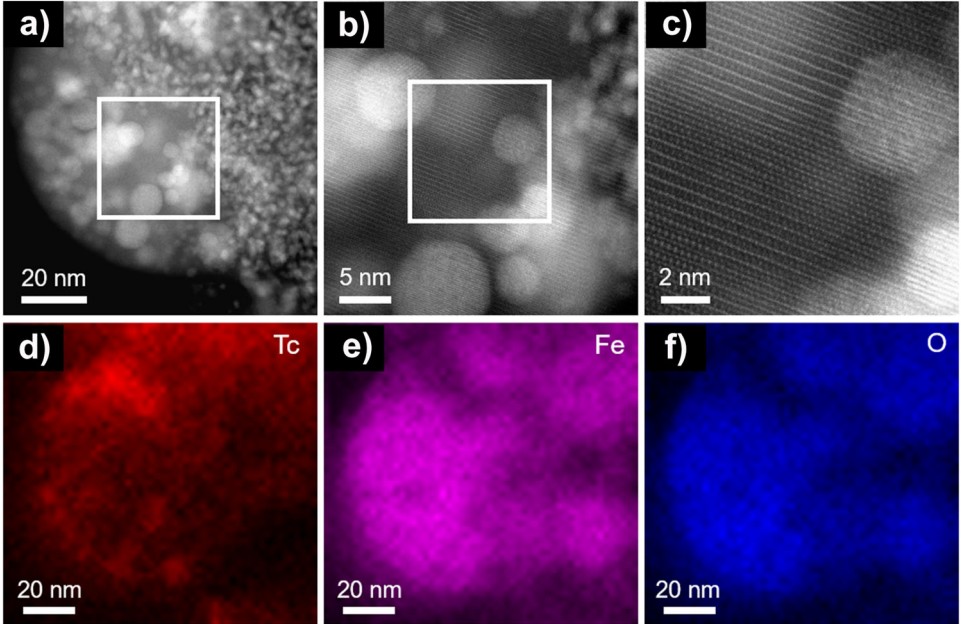

**Fig. 7 Microscopy analysis of the lamella at nano scale.** Granular ZVI reacted with 17 mM $TcO_4^-$ in 80 mM NaCl for one month. **a** HAADF STEM image of the sample; **b** higher magnification image of the area outlined in white on the image (**a**); **c** higher magnification image of the area outlined in white on the image (**b**), indicating the presence of magnetite or maghemite structure as seen along the [−1 1 2] zone axis; **d–f** EDS maps of Tc (**d**), Fe (**e**), and O (**f**).

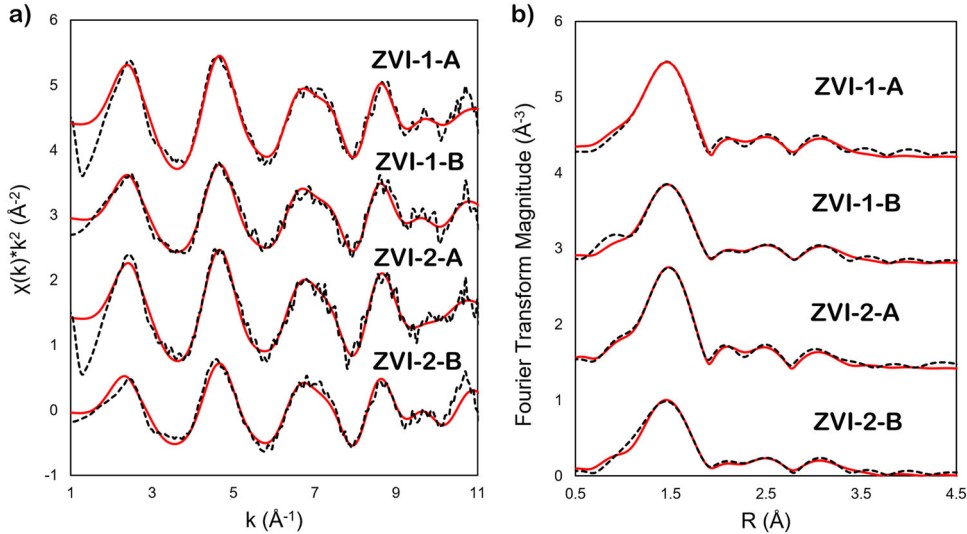

**Fig. 8 Extended X-ray absorption fine structure analysis.** ZVI granules reacted with 17 mM $TcO_4^-$ in 80 mM NaCl for one month: (**a**) $k^2$-weighted EXAFS spectra of Tc K-edge; (**b**) corresponding Fourier transform of $k^2$-weighted EXAFS; dotted black lines are data, solid red lines are fit; fitting k-range from 2 to 10 Å$^{-1}$; Fourier transform fitting range from 1.0 to 3.3 Å; amplitude reduction factor $S_0^2 = 0.8$.

with a higher fraction of $TcO_4^-$, i.e., 0.21 and 0.20, respectively (Table 1), the contribution from which is not included in the fit. The path Tc—Tc (Table 2) implies the existence of $Tc^{4+}$ oxide monomers and dimers[11]. Similar results were obtained for $TcO_2$ sorbed to magnetite[16], $Tc^{4+}$ partially incorporated into magnetite[18], and surface-precipitated $TcO_2 \cdot xH_2O$ (not incorporated into goethite, or magnetite[13]) with the respective Tc—Tc atomic distances at 2.57, 2.56, and 2.51 Å and CN of 0.9, 1.2, and 0.5.

The obtained Tc-Fe1 and Tc-Fe2 atomic distances (Table 2) are in excellent agreement with the respective 3.12 and 3.52 Å reported by Yalçintaş et al.[16]. These distances are very similar to those reported in the studies on sorbed and incorporated Tc in

magnetite[13,15,18,30], implying an impossible differentiation of the paths between sorbed vs. incorporated Tc in the magnetite structure. Note that the large value of $\sigma^2$ for the Tc-Fe1 path is an indication that the average CN of this path is lower, and is supportive of the heterogeneous system, where Tc has next-near neighboring Fe ions from the iron mineral, as well as Tc in a $Tc^{4+}$ oxide. Moreover, distances between iron atoms in different iron minerals are very similar, and the EXAFS spectra of hematite, goethite, and ferrihydrite were found to be almost identical[10,11]. In such cases conclusive statements should not be made from EXAFS analysis alone, but in conjunction with other techniques[20]. As such, STEM and EDS results confirm association of Tc with iron oxide, i.e., non-stoichiometric magnetite, not only on

**Table 2 Extended X-ray absorption fine structure fit parameters.**

| Scan location | Path | CN | R (Å) | σ² (Å²) | ΔE₀ (eV) | Chi_r² | R-factor |
|---|---|---|---|---|---|---|---|
| ZVI-1-A | Tc-O | 4.4 | 2.00 (2) | 0.004 (1) | $0.50 \pm 1.60$ | 419 | 0.010 |
| | Tc-Fe1 | 6.0[f] | 3.12 (5) | 0.022 (7) | | | |
| | Tc-Fe2 | 6.0[f] | 3.50 (4) | 0.012 (5) | | | |
| | Tc-Tc | 0.5 | 2.62 (4) | 0.003 (4) | | | |
| ZVI-1-B | Tc-O | 3.1 | 2.00 (1) | 0.002 (1) | $0.62 \pm 0.71$ | 33 | 0.008 |
| | Tc-Fe1 | 6.0[f] | 3.12 (2) | 0.020 (2) | | | |
| | Tc-Fe2 | 6.0[f] | 3.50 (1) | 0.011 (1) | | | |
| | Tc-Tc | 0.8 | 2.64 (2) | 0.007 (2) | | | |
| ZVI-2-A | Tc-O | 4.1 | 2.01 (2) | 0.003 (1) | $1.17 \pm 1.79$ | 143 | 0.013 |
| | Tc-Fe1 | 6.0[f] | 3.14 (7) | 0.024 (9) | | | |
| | Tc-Fe2 | 6.0[f] | 3.51 (6) | 0.013 (6) | | | |
| | Tc-Tc | 0.7 | 2.61 (4) | 0.003 (4) | | | |
| ZVI-2-B | Tc-O | 2.7 | 1.99 (1) | 0.001 (1) | $-1.10 \pm 1.16$ | 113 | 0.006 |
| | Tc-Fe1 | 6.0[f] | 3.09 (1) | 0.021 (3) | | | |
| | Tc-Fe2 | 6.0[f] | 3.48 (2) | 0.013 (3) | | | |
| | Tc-Tc | 0.8 | 2.64 (1) | 0.003 (2) | | | |

Fe1 refers to octahedral $Fe^{2+}$ or $Fe^{3+}$, and Fe2 is tetrahedral $Fe^{3+}$. Number in parenthesis is one SD error on the last significant figure; CN is coordination number; CN fit errors ±25%; [f]fixed R is atomic distance (Å); σ² is the EXAFS Debye-Waller factor (Å²); ΔE₀ is the shift in energy (eV); Chi_r² is the reduced Chi-square; R-factor is the closeness of fit. Fitting k-range from 2 to 10 Å⁻¹; Fourier transform fitting range from 1.0 to 3.3 Å. Amplitude reduction factor $S_0^2 = 0.8$.

the surface (Supplementary Fig. 6), but in the vertical profile of the lamella cut from the iron oxide with FIB technique (Figs. 6 and 7), showing a clustered Tc environment within the magnetite nanostructure. Possibly, the areas of local enrichment are mixtures of $TcO_2$ monomers and dimers incorporated in the iron oxide's crystal structure, where incorporation of mono-nuclear $Tc^{4+}$ at the octahedral Fe site cannot be excluded. This is also supported by the XPS results, demonstrating two distinct environments for $Tc^{4+}$. In addition, a model with contribution from the multiple scattering paths as an alternative to the iron shell scattering was tested and produced non-reportable results (Supplementary Table 3), supporting the evidence of incorporated $Tc^{4+}$.

## Discussion

Oxidation of $Fe^0$ in 80 mM NaCl in an aerobic environment resulted in near complete (99.8%) reductive removal of $Tc^{7+}$ from solution after almost a month from the start of the experiment. The sample showed discernible differences in the fractional distribution of iron minerals in comparison with the specimen oxidized under the same conditions but without Tc. The fraction of ferrihydrite, four times larger in the sample with Tc, was accompanied with a smaller fraction of non-stoichiometric magnetite, and a higher content of non-reacted ZVI. We relate these observations to the slower transformation of ferrihydrite to magnetite in the presence of Tc.

Transformation of ferrihydrite, a hydrous ferric oxyhydroxide with a low degree of order, commonly yields hematite and goethite (ferric oxide $\alpha$-$Fe_2O_3$ and oxyhydroxide $\alpha$-FeOOH, respectively), where predominant formation of one or the other is strongly dependent on pH and temperature, as well as on the degree of ordering of ferrihydrite and presence of anions, cations, and neutral molecules[2]. Adsorption of foreign species or their structural incorporation into iron mineral phases suppresses transformation processes; with higher concentrations of divalent metal ions, ferrihydrite transforms into goethite, mixtures of hematite and spinel, or spinel only when molar fraction of divalent cation 0.33 and higher[2,43]. High concentrations of ferrous ions ($Fe^{2+}$) induce a transformation path of ferrihydrite to magnetite (inverse spinel) at ambient temperature and pH >7 (faster at pH 9), which rapidly oxidizes to maghemite, as demonstrated in experiments with ferrihydrite and $Fe^{2+}$

solutions[41]. In our study, the initial amount of iron in the sample is high (50 g/L), leading to high $Fe^{2+}$ dissolution rates that are supported by pH 10.3 indicative of $Fe^0$ oxidation/dissolution reactions[2,27], implying that transformation of ferrihydrite to magnetite would prevail over hematite or goethite. The PXRD and Mössbauer spectral evidence reported here do not reveal a separate $Fe^{2+}$ mineral phase, e.g., siderite or green rust, that could have been expected due to the presence of carbonate in aerated solution at elevated pH. Furthermore, microscopic analysis of the iron oxidation products (Fig. 5) is very similar to that reported for spinel formation in $FeCl_2$ solution[41].

As mentioned previously, the addition of foreign species, i.e., anions, in the system retards the transformation of ferrihydrite, and our observations relate pertechnetate to the category of such species. The magnetite fraction was noticeably suppressed in the samples of ZVI with Tc (Supplementary Table 2), resulting in slower transformation of ferrihydrite into magnetite, which was also supported by PXRD data, with the amount of magnetite comprising 2–7% in the Tc-loaded samples vs. amount of magnetite in the range of 15–26% in the samples without Tc (Supplementary Fig. 3). Moreover, in the presence of Tc, samples of ZVI granules did not oxidize as fast, as evidenced by the larger amount of $Fe^0$ detected in the Tc-loaded samples via Mössbauer and PXRD analyses (Supplementary Table 2; Supplementary Fig. 3). These results are in accord with the fact that Tc is known for its anticorrosion properties[28,29].

Our experiments, carried out with initial Fe to Tc molar ratio 53, or 3.3 wt.% Tc, indicated the feasibility of near complete reduction of $Tc^{7+}$ ($TcO_4^-$ determined via XANES analysis could be explained by surface oxidation during sample handling and/or exposure to the photon beam) with consequent incorporation of at least 32% Tc, as determined from XANES analysis (Table 1) and supported by XPS analysis, which identified ~32% $Tc^{4+}$ in the local environment different from $Tc^{4+}$ in the precipitated oxide (Fig. 2). This amount comprises an estimated 1.86 wt.% Tc in the magnetite (magnetite fraction was estimated via Mössbauer measurements of the one-month reacted sample). Interestingly, these results, corresponding to the thermodynamically spontaneous system, support theoretical calculations defining 5 wt.% as the energetically feasible, but unstable, upper limit for Tc incorporation into magnetite[3] with increasing stability at 1.3 vs. 2.6 wt.% for Tc incorporation into hematite[12]. It is important to note that spinel ferrite with 4 wt.% Tc was synthesized by Lukens

et al.[17], however it pertains to a system under a controlled environment, i.e., $TcO_4^-$ dissolved in $Fe^{2+}$ solution followed by neutralization and heating. We emphasize the importance of a spontaneous system, analyzed here, as it relates to practical needs and may pave the way for remediation technologies and waste form development.

Further, our results showed heterogeneous distribution of Tc within the iron oxide (Fig. 7). Similar observations of clustered Tc in spinel ferrites have been reported previously for $Tc_{0.1}Fe_{2.9}O_4$[17], and $Tc^{4+}$ dimers were estimated to be the most stable clusters in doped rutile ($TiO_2$)[44]. The study by Yalçıntaş et al.[16]. reported $Tc^{4+}$ dimers bonded to the surface of pre-synthesized magnetite at 0.2 mM Tc as opposed to partial incorporation of $Tc^{4+}$ at 0.02 mM Tc (Fe:Tc molar ratios 1300–3200). It is known that substitution of magnetite is enhanced when the metal coprecipitates with ferrihydrite vs. when the metal is added after ferrihydrite precipitation[2]; that is why results for Tc incorporation are sensitive to relatively low Tc concentration of 0.2 mM[16], when there is no in situ synthesis of iron oxides/oxyhydroxides. In contrast, we observed significant fractional incorporation of $Tc^{4+}$ into iron oxide formed in situ during oxidation of $Fe^0$ at elevated Tc loading, 17 mM $TcO_4^-$ (Fe:Tc molar ratio 53). Such successful results can be explained by the compounded processes of $Tc^{7+}$ reduction and ferrihydrite formation with further transformation to non-stoichiometric magnetite or magnetite and maghemite mixture during oxidation of ZVI.

## Methods

**Caution**. $^{99}Tc$ emits $\beta^-$ radiation (0.29 MeV[29]). All radioactive samples were prepared at a nonreactor nuclear facility by certified and trained personnel inside of a fume hood designed for radiological contamination control.

**Batch experiments**. The samples were prepared with granular ZVI (Alfa Aesar, electrolytic, 1–2 mm particle size, 99.98% purity) at a concentration of 50 g/L in 80 mM NaCl (Sigma Aldrich, ≥99.5%); neutral solution (pH 6.9–7.3) using double deionized water (DIW) of at least 18 MΩ·cm. A $^{99}Tc$ stock, containing 40 g/L Tc as $NH_4TcO_4$, was used to prepare a working solution of 17 mM Tc in 80 mM NaCl which was added to granular ZVI samples (Fe to Tc molar ratio 53; or 3.3 wt.% Tc). All samples, with and without Tc, were placed on a shaking table for short-term (up to one week) and long-term (up to one month) contact times. Experiments were conducted at ambient temperature and pressure in the presence of air.

Concentrations of $Tc^{7+}$ in aqueous phase were determined by a liquid scintillation counter (Tri-Carb 3100TR, PerkinElmer) after samples were centrifuged and aliquots of 10–50 µL of solution were mixed with 10 mL of a scintillation cocktail (Ultima Gold LLT, PerkinElmer). Results were background corrected and accounted for the 94% efficiency of the counter.

**Solid phase characterization**. All samples, with and without Tc, were centrifuged to separate solids and decant the liquid phase. Separated solids were rinsed with DIW, removed by centrifugation, and dried under nitrogen atmosphere to prevent oxidation. Samples were preserved under a blanket of nitrogen until analyses.

Powder X-ray diffraction (PXRD) data were collected for all samples using a Rigaku Ultima IV diffractometer equipped with a Cu sealed tube X-ray source, using a divergent slit of 1/2°, a height limiting slit of 10 mm, 5° Soller slits on the source and detector sides, a 2/3° scattering slit and a 0.15 mm receiving slit, a bent graphite secondary monochromator and a scintillation detector. The diffractometer goniometer radius was 285 nm. Each PXRD sample was prepared in a poly(methyl methacrylate) plastic sample holder with a 25 mm diameter by 1 mm deep sample well and a dome cap. PXRD data of one month contacted samples were collected by sticking a small quantity of powder to a MiTeGen MicroMount pin with silicon vacuum grease. The pin mounted powder was placed on a Bruker D8 Venture single crystal diffractometer equipped with a Mo $K_\alpha$ X-ray source monochromated with a bent germanium crystal and a 2-dimensional area detector. The sample pin was rotated axially (φ rotation) during data collection. Diffraction frame data was integrated using the EVA version 4 (Bruker) software in order to generate an intensity vs. 2θ angle plot. Data were collected also for NIST SRM 660c in order to determine the zero error and refine parameters by method described by Cheary and Coelho[45] to account for instrument broadening. The diffraction data were analyzed using the TOPAS v6 software package. Identified phase components were quantified by the Rietveld method. Diffraction peak profile fitting was done using the double Voight method to account for peak broadening from crystallite size and micro-strain.

Mössbauer spectra were collected at room temperature for all samples, and, additionally, 77 K temperature analysis was used for the ZVI sample reacted with Tc for one month. The 50 mCi $^{57}Co/Rh$ source and velocity transducer MVT-1000 (WissEL) operated in a mode of constant acceleration (23 Hz, ±12 mm/s). The signal was transmitted through a holder where radiation was detected by Ar-Kr proportional counter. The counts were stored in a multichannel scalar as a function of energy, utilizing a 1024-channel analyzer. Data were folded to 512 channels to give a flat background and a zero-velocity position corresponding to the center shift (CS or δ) of a metal Fe foil at room temperature. A 25-µm thick Fe foil (Amersham, England) was placed in the same position as the samples to obtain calibration spectra. The Mössbauer data were modeled using the Recoil software (University of Ottawa, Canada) and a Voigt-based structural fitting routine.

X-ray photoelectron spectroscopy (XPS) spectra were recorded on a Kratos AXIS Ultra DLD system equipped with a monochromatic Al Kα X-ray source (1486.7 eV) at 10 mA, 15 kV for excitation and a hemispherical analyzer. Data were corrected relative to the reference 285.0 eV carbon 1 s peak. The software CasaXPS (version 2.3) with Shirley type background and 20% Gaussian–Lorentzian ratio was used for peak fitting.

X-ray absorption near edge structure (XANES) and extended x-ray absorption fine structure (EXAFS) measurements of the Tc K-edge were obtained at the X-ray Science Division beamline 20-ID-C at the Advanced Photon Source (APS, Argonne National Laboratory). In addition, 2-D elemental maps were collected based on x-ray fluorescence mapping (XRM). Thin sections were prepared for the analysis. Dried solids were prepared in a 1.6 cm (ID) aluminum tubing with slow drying epoxy. Two parts of epoxy were mixed with one part of hardener (EpoThin2 epoxy and hardener, Buehler, Lake Bluff, IL) with a volume of 3 mL with 0.1–0.5 g of solids. Samples were then vacuum-degassed to remove bubbles and air-dried overnight at room temperature. Slices were cut with an Isomet 1000 diamond blade thin sectioning saw with Isocut fluid (Buehler, Lake Bluff, IL) to a thickness of approximately 150 µm. After mounting on a glass microscope slide, thin sections were sanded with silicon carbide sand paper with Isocut fluid followed by 400 grit sand paper, then 600 grit sand paper, and finally 1200 grit sand paper (Ted Pella, Inc.) all using a figure eight pattern. Final polishing was conducted with a Nylon polishing cloth and 1 µm diamond polish (Metadi II, Buehler). Samples were washed with methanol and dried prior to analysis. Samples of ZVI granules contacted with Tc were encased in epoxy resin and cut into thin sections. The sample was doubly encapsulated in mylar and Kapton films.

Spectra were collected in fluorescence mode using a 4 element Vortex (silicon drift) detector, Si 111 monochromator; Fe and Mo foils were utilized for energy calibration. Data were analyzed using ATHENA and ARTEMIS software (version 0.9.26)[46]. XANES spectra were analyzed via linear combination fitting with standards of $Tc^{4+}$ in magnetite[17], $Tc^{4+}$ oxide[47] and $Tc^{7+}$ adsorbed onto Purolite A530E resin. ICSD files of $Fe_3O_4$ and $TcO_2$ were used for FEFF6.0 calculations for fitting the EXAFS region; the magnetite input file was modified with a core Tc atom in an octahedral Fe site.

Scanning electron microscopy (SEM) images were obtained using a Quanta 250FEG SEM (Thermo-Fisher Inc., Hillsboro, OR) equipped with an EDAX Genesis™ (EDAX Inc., Mahwah, NJ) analytical energy dispersive X-ray spectroscopy (EDS) system. Scanning transmission electron microscopy (STEM) was performed on a cold field emission JEOL ARM 200 F operated at 200 kV and equipped with a Noran™ (Thermo Scientific, Waltham, MA) EDS system. The microscope is aberration-corrected with a hexapole-type probe Cs corrector (CESCORR, CEOS). STEM micrographs and EDS maps were collected with a 0.78-Å probe at an emission current of 15 µA, with a high-angle annular dark-field (HAADF) detector inner angle of 82.6 mrad. A lamella from the sample contacted with Tc for one month was prepared using a focus ion beam (FIB) on a Quanta SEM (Thermo Fisher FEI). The lamella was attached to a copper Omniprobe liftout grid using Pt welds. The sample aged without Tc produced loosely aggregated clumps of iron oxides and several lamellae from that sample fragmented before imaging. Therefore, the Tc-free sample was prepared for imaging by directly placing the dry particles onto a 200-mesh copper TEM grid coated with a holey carbon film (Electron Microscopy Supplies, Inc) and tapping off the excess.

## Data availability
Data supporting these findings are available within the article and supplementary information files.

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

## Acknowledgements

It is our pleasure to acknowledge help of C. T. Resch, S. Chatterjee, M. Fujimoto, W. W. Lukens, C. I. Pearce, K. Kruska, M. J. Olszta, R. G. Surbella, and D. L. Brewe. This research was supported by the U.S. Department of Energy's Office of Environmental Management and performed as part of the Technetium Management Hanford Site project at the Pacific Northwest National Laboratory (PNNL) operated by Battelle for the U.S. Department of Energy under Contract No. DE-AC05–76RL01830. A portion of the microscopy was performed in the Radiological Microscopy Suite, located in the Radiochemical Processing Laboratory at PNNL. This work was in part supported by the Department of Energy Minority Serving Institution Partnership Program (MSIPP) managed by the Savannah River National Laboratory under SRNS contract DE-AC09-08SR22470. This research used resources of the Advanced Photon Source, a U.S. Department of Energy (DOE) Office of Science User Facility operated for the DOE Office of Science by Argonne National Laboratory under Contract No. DE-AC02-06CH11357. We also acknowledge '2019 APS/IIT XAFS Summer School' in Chicago, IL.

## Author contributions

T.G.L. directed overall research, designed experiments, and guided manuscript preparation. Y.K. and H.P.E. supervised the study. J.A.S. carried out STEM/EDS analysis; R.K.K. collected Mössbauer data and performed modeling; Y.D. carried out XPS analysis with modeling; L.E.S. carried out PXRD analysis with modeling; E.C.B. collected SEM data; V.E.H. collected XAFS data at APS; C.U.S. contributed to EXAFS data modeling; G.B.H. prepared samples for analyses; D.B. executed experiments, analyzed data, and wrote the manuscript. All the authors revised the manuscript.

## Competing interests

The authors declare no competing interests.
