## [Peer Review File · Communications Chemistry]

In their review of the first and second version of this manuscript, reviewer 2 added their comments to the manuscript file.

These comments have been added as an attachment in this Peer Review File.

Reviewers' comments:

Reviewer #1 (Remarks to the Author):

The manuscript by Boglaienko et al. reports the reduction of Tc(VII) to Tc(IV) by ZVI, with the resulting Tc(IV) being strongly partitioned to the solid phase. The authors claim that Tc(IV) is incorporated into the structure of magnetite (the phase produced through oxidation of ZVI) at a loading of 1.86 wt%.

Whilst I find the work interesting and a good, novel avenue for exploration, I cannot recommend this manuscript for publication. Most importantly, I feel the data, whilst good quality, has been fundamentally misinterpreted meaning that the conclusions drawn are incorrect (in particular the XANES and EXAFS analysis of the Tc phases present and the iron mineral transformation pathways). Additionally, I find some of manuscript quite difficult to follow and unclear.

I believe that the work could be improved through a reconsideration of the presented data, the addition of some fundamental geochemical data (such as aqueous Tc and Fe plots and pH profiles), and rewriting sections of the text to provide clarity and demonstrate the uncertainties around the results. My recommendation would be for the authors to significantly revise the manuscript with additional data and reanalysis and I wish them well in publishing this manuscript in a more interest specific area.

My full review comments are provided in the uploaded document.

Reviewer #2 (Remarks to the Author):

The paper comprises excellent and very rich analytical work with bad control of experimental conditions. Therefore I suggest to reject the paper. Detailed comments are included in the annotated manuscript added to the review. In contrast to the claim of the authors, it is not the first time that remediation of Tc containing environments by ZVI is proposed. I added some references in the annotated paper. The initial Tc/Fe ratio is unrealistically high for any remediation issues, governed by analytical needs. But the speciation of Tc in the Fe will change with the Tc/Fe ratio and the clustering will certainly be less important in a real case. The authors also invoked the usefulness of the information of the article for waste form fabrication, but this is certainly not a good strategy to produce wastefoms under such uncontrolled conditions. Indeed, the largest drawback of the article is that it is focussed on a highly dynamic system in which the Fe(0) is oxidized under conditions which are not described, an will not only depend on time but as well on the air access to the samples, of stirring rates etc. This dynamic system has been studied without studing the evolution as a function of time. The paper contains also quite a bit of speculation, as for example to the formation of dimeric Tc(IV). While it is clear that clusters are observed, no indications for dimers are provided. The incorporation of Tc in iron corrosion products is clearly observed, but it is not clear to which degree solid solutions are formed and to which degree simple entrapment of clusters of unknown size in the precipitating FeII/III phases.

Reviewer #3 (Remarks to the Author):

This paper describes the phenomena associated with the interaction of pertechnetate with metal iron and the change of oxidation states that follow. It is a well-written document reporting the work of a highly diverse team of scientists on multiple techniques.

The major claim of the paper is the quantitative reduction of Tc(VII) by iron and oxidation of iron to magnetite.

This paper provide a substantial advance in science and in the field of nuclear waste management.

Overall, this is a high-quality paper that has its place in Communications Chemistry.

I have a few comments:

Line 99-100: The authors point out that the chloride ion will not influence the system because of Cl^- inability to complex. While chloride complexation with cation is minimal, it is not entirely inexistent. Moreover, the designation of " Tc^{7+} " is misleading, as heptavalent Tc is in the form of the anionic species TcO_4^- and does not act as a cation. Finally, speaking of counter anion of concern, the authors should discuss the influence of the presence of NH_4^+ , present in the system at the same concentration as Tc, at 17 mM.

Figure 2, line 242: I am surprised the subsequent magnetic shows so much crystallinity. Under such mild condition of pressure, one should expect an amorphous solid. I would like to see the author providing some insight on the matter.

Table 1: I find the errors provided uncomfortably low.

General Comments

The manuscript by Boglajenko et al. reports the reduction of Tc(VII) to Tc(IV) by ZVI, with the resulting Tc(IV) being strongly partitioned to the solid phase. The authors claim that Tc(IV) is incorporated into the structure of magnetite (the phase produced through oxidation of ZVI) at a loading of 1.86 wt%. Whilst I find the work interesting and a good, novel avenue for exploration, I cannot recommend this manuscript for publication. Most importantly, I feel the data, whilst good quality, has been fundamentally misinterpreted meaning that the conclusions drawn are incorrect (in particular the XANES and EXAFS analysis of the Tc phases present and the iron mineral transformation pathways). Additionally, I find some of manuscript quite difficult to follow and unclear. I believe that the work could be improved through a reconsideration of the presented data, the addition of some fundamental geochemical data (such as aqueous Tc and Fe plots and pH profiles), and rewriting sections of the text to provide clarity and demonstrate the uncertainties around the results. My recommendation would be for the authors to significantly revise the manuscript with additional data and reanalysis and I wish them well in publishing this manuscript in a more interest specific area.

Introduction Comments

Generally, I feel the introduction is lacking somewhat in background of ZVI and the expected mineral transformations that take place under aerobic and anaerobic conditions. As I note later, the general mineral transformation pathways proposed in the manuscript aren't clear to me and having a short paragraph on the background of what mineral transformations and end members are expected would be beneficial for providing context in the manuscript and framing the arguments presented in the manuscript.

Methods Comments

Was the deionised water used aerobic or degassed? This may cause issues with oxidation (particularly possible surface oxidation that the authors themselves noted in the manuscript (see lines 117-118, 121-122, and 353)) when washing the samples for solid characterization as stated. This may explain why there was an increase in Tc concentration in the washing solution compared to the supernatant.

Further to this, solid characterization is stated to be done via centrifugation to separate solids from the supernatant. However, I'm confused by a statement in the 'Results' section (lines 113-114), that seems to suggest that there are considerable mineral fines (postulated to be magnetite) associated with the aqueous phase and it is not clear if these are removed to the solid phase by centrifugation or not. If these mineral phases weren't removed by centrifugation then there has not been a true solid/solution separation which impacts other results in the manuscript.

Results Comments

A general point about the results is I'm unsure why there is an absence of basic aqueous geochemical data in this section. Whilst some numbers are provided for Tc concentration in this section (but pH values are only stated in the 'Discussion' section?), why wasn't aqueous Fe in solution measured by ICP-MS or a ferrozine assay performed to quantify the Fe(II)/Fe(III) ratio? This could yield important information on mineral transformations taking place. Additionally, plots of Tc conc and pH changes over time (even just in the SI) would make interpreting the data much easier.

I'm uncertain of what is meant in lines 116-118 regarding the concentration of Tc in the supernatant as the concentration quoted (80 mM) is higher than the initial amount of Tc added. I assume this is

because the washing volume is smaller than the supernatant and the reaction mixture but this isn't clear and I think using an amount may give more context to the point being made here.

There are some points around the Linear Combination Fitting (LCF) of the XANES spectra that I think need to be addressed. Firstly, a minor editorial point but lines 122-123 begin discussing a 'fit' without prior discussion of LCF being performed. A statement of clarity should be added in here.

Linear combination fitting of XANES spectra is somewhat of a crude process and requires thorough discussion and requires a thorough step-wise approach in order to gain any meaningful understanding of the data. Additionally, performing with LCF with more than two standards can sometimes be seen as problematic due to how approximate the process is. For more definite numbers of the proportions of species present, ITFA analysis should be performed. However, a more thorough presentation of other LCF fitting data would also be useful in order to determine whether the 3 standard fits and the corresponding numbers are meaningful. LCF fits of all possible combinations of two standards should be provided in order to determine whether or not the presence of 3 standards is needed as qualitatively (and from the fitting parameters provided), the XANES don't seem to have much of a Tc(VII) component and look similar to the TcO₂ standard. F-testing the addition of each standard may also be useful to prove that the addition of each standard to the fit is statistically valid.

Is the fitting window for the LCF the same for all spectra? I assume so but it might be useful to add a statement of clarity into the Table 1 caption (as the authors have done with Table 2).

Also, LCF of the EXAFS could be performed and a comparison between the XANES and EXAFS LCF fits could give insight into the components present in the solid phase.

With respect to XPS, why was the Fe 2p spectrum not fit using CASA XPS? This could give you information on the Fe(II)/Fe(III) ratio in the solid phase and shed light on the possible mineral phase present. The XPS data itself looks much more like an Fe(III) mineral when compared with literature data (Radu et al., Applied Surface Science, 2017) and whilst I realise that this is somewhat addressed in the statement in lines 158-160, the results may be somewhat valid of the bulk is the sample is indeed 'nano-sized' as the authors state in line 192.

The PXRD data casts doubt on the exact mineral phase present as the authors correctly state that magnetite and maghemite cannot be discriminated between with this technique. Furthermore, the Mössbauer spectroscopy seems to strengthen the case for mineral phase uncertainty. Whilst I appreciate the authors have stated in lines 204-206 that they refer to magnetite as being 'non-stoichiometric' due to the uncertainty, I don't believe that this is necessarily carried on throughout the manuscript. Furthermore, I don't believe the level of uncertainty surrounding the mineral phase present is conveyed clearly (for example in the Abstract) as a lot of data (including microscopy as the morphology of magnetite and maghemite are very similar under SEM & TEM, for example, Legodi & de Waal, Dyes & Pigments, 2007) does not definitely suggest magnetite being present any more than an Fe(III) phase like maghemite.

Lines 246-247 regarding the mineral transformation pathway are confusing as there is no context for the expected or proposed transformation taking place here. Clarity through extra background or moving this statement to the discussion may help.

With respect to the EXAFS analysis section, I'm unsure of why this data and the XANES are punctuated by the Fe mineral phase characterisation as it makes the manuscript difficult to follow.

Upon inspection of the data, I fundamentally disagree with the conclusions drawn from this and I do believe these spectra show Tc incorporated into an iron oxide mineral phase. I have provided greater

detail on some of the specifics of the data analysis below, however, a comparison with previous published literature (Marshall et al., Environmental Science & Technology, 2014) clearly shows that the spectra provided in this study are not incorporated Tc species and suggests far more strongly that the phase present here is indeed TcO₂ with no Fe association. This can be seen clearly by the lack of large Fe peaks in the Fourier transform between 3 and 4 Å that are shown in Marshall et al. (2014) and that indicate Tc incorporation.

Additionally, why are the EXAFS (k-space) presented in k² weighting when the convention is k³? This should be changed to make comparisons with other literature EXAFS data. Additionally, Table 2 (and Table 1 for that matter) doesn't need the reduced chi squared column as the number is meaningless unless comparing between fits of the same dataset (<https://bruceravel.github.io/demeter/documents/Athena/analysis/lcf.html> – the link provided is for LCF fitting but is true for EXAFS fitting too).

The coordination numbers for the first Tc-O shell are much lower than the anticipated 6 fold coordination of Tc in a Tc(IV) oxidation state. Could the authors comment on why they believe this is the case?

The Debye-Waller factors for the first Fe shell (Tc-Fe1) in all samples, whilst not necessarily incorrect, seems to be quite large. Especially after consulting the literature (including the references provided in the manuscript) as other reported Debye-Waller factors are not this high. This could mean that this shell isn't really contributing to the fit and this shell should be F-tested to prove it is needed for a good fit. Whilst it may be convention to fix the Fe shells to a CN of 6, reducing these slightly may result in better fits and better fitting parameters. Further to this, if the samples in any way nanocrystalline or disordered, a decrease in CN of more distant shells would be expected.

Discussion Comments

Is the pH stated (10.3) the final, stable pH? How did pH change throughout the experiment? Has the system reached equilibrium or could there still be transformations taking place?

The discussion states '[the] the high dissolution rate of Fe²⁺ was expected to induce transformation of ferrihydrite into magnetite...' but how do you know this when there is no aqueous geochemical data showing Fe dissolution?

Could the authors provide some clarity on how they determined the wt% Tc incorporated into magnetite as I am uncertain of how the value of 1.86 wt% is calculated. If, as the authors state, there is only 2-7 % magnetite in their system (according to PXRD), this would mean that of the 50 g/L ZVI added to the system, only 3.45 g/L are magnetite if take an approximate mean abundance of 5 % magnetite. Considering the authors state that approximately 32 % of the 17 mM Tc added to the experiment was incorporated this would mean 0.5 g/L Tc would need to be incorporated into 3.45 g/L of magnetite and the incorporation would be ~14 wt%. This value is much higher than previously reported values and likely indicates that the incorporated proportion of Tc into magnetite is incorrect.

The discussion needs more of clarity around the overall proposed pathways and mechanisms taking place here. Not just for the mineral transformations, but the pathways and fate of Tc. A concise, clear conclusions paragraph would achieve this.

General Editorial Comments

Line 38 – ‘Further, formation of iron minerals is of singular complexity’ – I’m not overly clear what this statement means.

Line 89 – ‘routs’ should be replaced with ‘routes’ if I’m inferring the statement correctly.

Lines 291-292 – ‘junction’ should be replaced with ‘conjunction’ if I understand the sentence correctly.

Line 339 – ‘, Fig. 4,’ is inconsistent and should be changed to ‘(Fig. 4)’.

Response to Reviewers' comments:

TITLE: Spontaneous Tc⁷⁺ – Fe⁰ Redox Continuum: Sequestered Tc⁴⁺ Clusters and Retarded Mineral Transformation of Iron
Communications Chemistry

Reviewer 1:

General Comments

The manuscript by Boglaienko et al. reports the reduction of Tc(VII) to Tc(IV) by ZVI, with the resulting Tc(IV) being strongly partitioned to the solid phase. The authors claim that Tc(IV) is incorporated into the structure of magnetite (the phase produced through oxidation of ZVI) at a loading of 1.86 wt%. Whilst I find the work interesting and a good, novel avenue for exploration, I cannot recommend this manuscript for publication. Most importantly, I feel the data, whilst good quality, has been fundamentally misinterpreted meaning that the conclusions drawn are incorrect (in particular the XANES and EXAFS analysis of the Tc phases present and the iron mineral transformation pathways). Additionally, I find some of manuscript quite difficult to follow and unclear. I believe that the work could be improved through a reconsideration of the presented data, the addition of some fundamental geochemical data (such as aqueous Tc and Fe plots and pH profiles), and rewriting sections of the text to provide clarity and demonstrate the uncertainties around the results. My recommendation would be for the authors to significantly revise the manuscript with additional data and reanalysis and I wish them well in publishing this manuscript in a more interest specific area.

Response

We greatly appreciate the significant amount of time the reviewer has put into the thorough and in-depth review. We will also try our best to address each comment (they were numbered for convenience) accurately and thoughtfully. For every comment, a corresponding change is marked in red in the manuscript draft, and, in case of clarification and justification, every statement is attempted to be carefully explained.

Introduction Comments

Comment 1

Generally, I feel the introduction is lacking somewhat in background of ZVI and the expected mineral transformations that take place under aerobic and anaerobic conditions. As I note later, the general mineral transformation pathways proposed in the manuscript aren't clear to me and having a short paragraph on the background of what mineral transformations and end members are expected would be beneficial for providing context in the manuscript and framing the arguments presented in the manuscript.

Response:

Iron mineral transformations take many different routes, depending on numerous factors (solution composition, pH, dissolved oxygen, iron surface passivation and dissolution rates of iron, etc.), in each particular system it might take different pathway, and it precludes generalized description of the iron oxidative pathway. The main transformation paths that apply to our

system are given in Discussion (pages 15-16, second paragraph), where we think they are more needed, because they can be crosslinked with discussion of our data.

Page 2, a sentence was added to the Introduction section:

“Iron oxidation in anoxic vs oxic conditions would take a different route with respective water or dissolved oxygen serving as an oxidant, leading to dissimilar iron oxidation pathways.²”

Methods Comments

Comment 2

Was the deionised water used aerobic or degassed? This may cause issues with oxidation (particularly possible surface oxidation that the authors themselves noted in the manuscript (see lines 117-118, 121-122, and 353)) when washing the samples for solid characterization as stated. This may explain why there was an increase in Tc concentration in the washing solution compared to the supernatant.

Response:

The objective of this was to understand the behavior of the system and Fe(0) mineral transformation in an aerobic environment as it is stated in the Introduction, lines 73-74. Therefore, the deionized water was aerobic, as the experimental series were conducted using aerobic solutions.

To avoid possible uncertainty, the Abstract is modified as follows: “Incorporation of Tc⁴⁺ into iron minerals has been studied predominantly for systems under conditions of carefully controlled **anaerobic** environment, employing syntheses with Fe²⁺ solutions or using pre-synthesized iron oxides/oxyhydroxides, and mechanisms of the transformation of iron phases leading to incorporation of Tc⁴⁺ **under aerobic conditions** remain poorly understood.”

Comment 3

Further to this, solid characterization is stated to be done via centrifugation to separate solids from the supernatant. However, I’m confused by a statement in the ‘Results’ section (lines 113-114), that seems to suggest that there are considerable mineral fines (postulated to be magnetite) associated with the aqueous phase and it is not clear if these are removed to the solid phase by centrifugation or not. If these mineral phases weren’t removed by centrifugation then there has not been a true solid/solution separation which impacts other results in the manuscript.

Response:

Yes, centrifugation removed these mineral fines from the aqueous phase, and separated bulk solid phase was analyzed for each sample. Centrifugation was repeated several consequent times to achieve separation of fine solids.

Page 3, Results section, a sentence was added:

“... **and analyzed as a bulk solid phase with the rest of solids per each sample**”

Results Comments

Comment 4

A general point about the results is I'm unsure why there is an absence of basic aqueous geochemical data in this section. Whilst some numbers are provided for Tc concentration in this section (but pH values are only stated in the 'Discussion' section?), why wasn't aqueous Fe in solution measured by ICP-MS or a ferrozine assay performed to quantify the Fe(II)/Fe(III) ratio? This could yield important information on mineral transformations taking place. Additionally, plots of Tc conc and pH changes over time (even just in the SI) would make interpreting the data much easier.

Response

The study presented in the manuscript is the continuation of the previous extensive work that has been done with ten commercially available ZVI materials, including granular ZVI used in the current manuscript, for which the pH, ORP, Fe(II) and Fe(III) dissolution and other analyses were conducted along with the Tc reduction kinetics up to 30 days [Boglaienko, et al. *J. Hazard. Mater.* **380**, 120836 (2019)]. Dissolved iron was measured by the by ultraviolet-visible Ferrozine method (at 562 nm wavelength) with the sampling time steps 10 min, 30 min, 1 hr, 3 hrs, 1 day, 2 days, 5 days and 8 days. The solution matrix was the same as in the current work, 80 mM NaCl, very close around neutral initial pH (in the range of 6.9 – 7.3), aerobic environment and normal conditions. Among those tested materials, we chose granular ZVI for the further experiments, which composed current manuscript. The choice of this material is mentioned in the last paragraph of Introduction (page 3) and the reference to the previous study, which has pH, ORP, Tc kinetics data was also given [27]. We added additional sentences after the first paragraph of the Results section, clarifying that missing data have already been presented in the previous work.

Page 4, Results section, a paragraph was added:

As it was investigated previously, ZVI materials manufacturing by different methods exhibit different kinetic of Tc^{7+} removal attributed to variable surface properties and rates of Fe^0 oxidation and dissolution²⁷; among the wide variety of the iron materials tested before, granular iron belonged to a ZVI group of the superior Tc^{7+} reduction efficiency and exhibited the least propensity to Tc^{4+} re-oxidation within this group. For this material, profile of the Fe^0 solubilization in 80 mM NaCl solution and the corresponding pH and ORP time dependencies had been studied earlier.²⁷ Here, the pH value of 10.3 measured 25 days after the start of experiment indicated on prevalence of the iron dissolution reactions.²

Comment 5

I'm uncertain of what is meant in lines 116-118 regarding the concentration of Tc in the supernatant as the concentration quoted (80 mM) is higher than the initial amount of Tc added. I assume this is because the washing volume is smaller than the supernatant and the reaction mixture but this isn't clear and I think using an amount may give more context to the point being made here.

Response

The volume of the washing solution (deionized water) was approximately the same as the initial amount of the contact solution (10 mL), meaning that we can compare concentrations of Tc in

the supernatant and the washing solution. There could have been resuspension of the Tc^{4+} precipitated to the bottom in the form of oxide or re-oxidation of Tc^{4+} to Tc^{7+} .

Page 3: We deleted the phrase “due to the presence of $\text{TcO}_2 \cdot n\text{H}_2\text{O}$ ” as in the logical sequence of the results narrative, the form of Tc has not been defined yet.

Comment 6

There are some points around the Linear Combination Fitting (LCF) of the XANES spectra that I think need to be addressed. Firstly, a minor editorial point but lines 122-123 begin discussing a ‘fit’ without prior discussion of LCF being performed. A statement of clarity should be added in here.

Response

Page 4: a sentence was added:

“Linear combination fit (LCF) was performed for XANES and EXAFS spectra...”

Comment 7

Linear combination fitting of XANES spectra is somewhat of a crude process and requires thorough discussion and requires a thorough step-wise approach in order gain any meaningful understanding of the data. Additionally, performing with LCF with more than two standards can sometimes be seen as problematic due to how approximate the process is. For more definite numbers of the proportions of species present, ITFA analysis should be performed. However, a more thorough presentation of other LCF fitting data would also be useful in order to determine whether the 3 standard fits and the corresponding numbers are meaningful. LCF fits of all possible combinations of two standards should be provided in order to determine whether or not the presence of 3 standards is needed as qualitatively (and from the fitting parameters provided), the XANES don't seem to have much of a Tc(VII) component and look similar to the TcO_2 standard. F-testing the addition of each standard may also be useful to prove that the addition of each standard to the fit is statistically valid.

Response

In support of the LCF, we attempted to check the number of principal components (PCA) in the collected Tc spectra. Even though we did and compared PCA for the whole spectrum, XANES and EXAFS regions, the focus was placed on EXAFS region (k-range from 3 to 12), because spectra of Tc with the same oxidation state but in different local environments would exhibit more variability in EXAFS region than in XANES. Reconstruction of spectra for EXAFS region supports three components:

The following reconstruction graphs show almost no residual noise in the 3 components graph (on your right). Residual noise in the graph with two-component reconstruction on your left shows sine-like signal that might not be discarded.

In addition, LCF combinatorics were compared for EXAFS ($k = 3$ to 12), and the results are following:

Standards: A – Tc in Fe3O4 B – TcO2 nH2O C – TcO4	R-factor	Reduced chi 2
A, B, C	0.38796	0.0472
A, B	0.63044	0.0763
B, C	0.65905	0.0798
A, C	0.81133	0.0983

The R-factor and reduced chi 2 are almost twice smaller for three components vs two components.

More explanation and support of the conclusions that Tc existed in two oxidation states and three local environments are given later in this response.

Page 6, Table 1: LCF results were appended with LCF EXAFS (k -range from 3 to 12), and R-factor was excluded. We kept reduced chi square as a statistics parameter.

Page 4: sentences were added in Results section:

“Linear combination fit (LCF) was performed for XANES and EXAFS spectra, and LCF combinatorics showed better statistical results for three components, with R-factor and χ_r^2 almost twice smaller for three components vs two components. Therefore, three standards were kept for LCF”

Comment 8

Is the fitting window for the LCF the same for all spectra? I assume so but it might be useful to add a statement of clarity into the Table 1 caption (as the authors have done with Table 2).

Response

Yes, it was the same for all the spectra and corresponds to the plotted range (Fig. 1).

Page 6: Title of Table 1 was appended:

“Linear combination fitting results for XANES (21 to 21.4 keV) and EXAFS data ($3 < k < 12$).”

Comment 9

Also, LCF of the EXAFS could be performed and a comparison between the XANES and EXAFS LCF fits could give insight into the components present in the solid phase.

Response

LCF was performed for EXAFS and results are presented in the Table 1:

	XANES Weight fraction (one SD error on the last significant figure)	XANES χ_r^2	EXAFS χ_r^2 Weight fraction (one SD on the Last significant figure)	EXAFS
ZVI-1-A				
TcO ₄ ⁻	0.11 (1)		0.10 (1)	
TcO ₂ •nH ₂ O	0.55 (6)	0.009	0.48 (3)	0.04
Tc ⁴⁺ in Fe ₃ O ₄	0.34 (6)		0.42 (4)	
ZVI-1-B				
TcO ₄ ⁻	0.21 (2)		0.15 (2)	
TcO ₂ •nH ₂ O	0.45 (9)	0.021	0.45 (4)	0.05
Tc ⁴⁺ in Fe ₃ O ₄	0.34 (9)		0.40 (4)	
ZVI-2-A				
TcO ₄ ⁻	0.13 (1)		0.10 (1)	
TcO ₂ •nH ₂ O	0.55 (6)	0.010	0.52 (3)	0.03
Tc ⁴⁺ in Fe ₃ O ₄	0.32 (6)		0.39 (3)	
ZVI-2-B				
TcO ₄ ⁻	0.20 (2)		0.15 (1)	
TcO ₂ •nH ₂ O	0.46 (9)	0.020	0.47 (3)	0.05
Tc ⁴⁺ in Fe ₃ O ₄	0.34 (9)		0.38 (4)	

As it can be seen from the table 1, EXAFS LCF results are very consistent with LCF results for XANES supporting validity of the analysis.

Comment 10

With respect to XPS, why was the Fe 2p spectrum not fit using CASA XPS? This could give you information on the Fe(II)/Fe(III) ratio in the solid phase and shed light on the possible mineral phase present. The XPS data itself looks much more like an Fe(III) mineral when compared with literature data (Radu et al., Applied Surface Science, 2017) and whilst I realize that this is somewhat addressed in the statement in lines 158-160, the results may be somewhat valid of the bulk is the sample is indeed 'nano-sized' as the authors state in line 192.

Response

We considered fitting iron data, but in our case it can lead to erroneous interpretation. The core-level XPS Tc 3d spectrum is composed of Tc 3d_{5/2} and 3d_{3/2} resulting from spin orbit splitting, and requiring doublets for peak fitting. In Fig. 2 we peak fit Tc 3d using three pairs of well-defined doublets, each corresponding to a different oxidation state, or chemical environment. However, Fe 2p spectrum is much more complicated to fit. The Fe 2p spectrum from pure Fe(III) compounds display complex multiplet-split due to unpaired electrons and satellite features with one being in between Fe2p_{3/2} and Fe2p_{1/2} (Surf. Interface Anal. 2004, 36, 1564). When a core electron vacancy is created by photoionization, there can be coupling between the unpaired electron in the core with the unpaired electrons in the outer shell. This can create a number of final states, which leads to that fitting Fe2p_{3/2} spectrum of Fe(III) requires a multi-peak envelope. Similarly, Fe(II) spectrum in FeO (high-spin) also exhibits complex multiple splitting with different satellite features. Due to the complex multiplet-splitting of Fe containing compounds, the peak fitting approach can often lead to erroneous assignment. On the other hand, the satellite feature for Fe(II) in between Fe2p_{3/2} and Fe2p_{1/2} has a lower binding energy compared to that of Fe(III), which has often been used to distinguish the valence state of Fe as shown in Fig. 42, 63, and 78 in Surface Science Reports, 2016, 71, 272.

Comment 11

The PXRD data casts doubt on the exact mineral phase present as the authors correctly state that magnetite and maghemite cannot be discriminated between with this technique. Furthermore, the Mössbauer spectroscopy seems to strengthen the case for mineral phase uncertainty. Whilst I appreciate the authors have stated in lines 204-206 that they refer to magnetite as being ‘nonstoichiometric’ due to the uncertainty, I don’t believe that this is necessarily carried on throughout the manuscript. Furthermore, I don’t believe the level of uncertainty surrounding the mineral phase present is conveyed clearly (for example in the Abstract) as a lot of data (including microscopy as the morphology of magnetite and maghemite are very similar under SEM & TEM, for example, Legodi & de Waal, Dyes & Pigments, 2007) does not definitely suggest magnetite being present any more than an Fe(III) phase like maghemite.

Response

In the given system there can be neither pure form of magnetite, nor maghemite, but transition between these two during partial oxidation of magnetite, however not yet complete transformation to maghemite. In this case we relied on the Mossbauer data mainly, i.e. calculation of the magnetite ratio that allows to see the transformation levels of the magnetite. Ratio was calculated according to Gorski & Scherer *Am. Mineral.* (2010) and explanations of the differences between two minerals are given in the sub-section Characterization of ZVI solid phase oxidized in the presence and absence of Tc, while the ratios are compared in Supplementary Table 1 in SI.

Clarifications were added throughout the manuscript.

Abstract: “Quantitative reduction of TcO₄⁻ and structural incorporation of Tc⁴⁺ into **non-stoichiometric** magnetite benefited from concomitant Fe⁰ oxidative transformation”

Page 13, EXAFS analysis of Tc K-edge in ZVI granules: “Even though non-stoichiometric magnetite or magnetite and maghemite mixture is referred to here, due to their similarities, a modified crystal structure of magnetite was used for the fitting model.” and “... STEM and EDS results confirm association of Tc with iron oxide, *i.e.* non-stoichiometric magnetite...”

Page 17, last conclusive sentence in the Discussion section: “...further transformation to non-stoichiometric magnetite or magnetite and maghemite mixture during oxidation of ZVI”

Comment 12

Lines 246-247 regarding the mineral transformation pathway are confusing as there is no context for the expected or proposed transformation taking place here. Clarity through extra background or moving this statement to the discussion may help.

Response

Discussion section is mainly focused on description of iron transformation pathways (starting from ferrihydrite) and, due to limitation of space, we could not repeat it several times. However, it was important to mention ferrihydrite in this place, as it was already defined via PXRD in the preceding sub-section: “In addition, single crystal diffractogram of the one month contacted ZVI sample without Tc revealed additional iron phase, ferrihydrite (Supplementary Fig. 4)”. We think, it is important to leave it mentioned here, as it connects this thought lines throughout the manuscript.

Comment 13

With respect to the EXAFS analysis section, I’m unsure of why this data and the XANES are punctuated by the Fe mineral phase characterisation as it makes the manuscript difficult to follow.

Response

We agree that it is odd to break these analyses, as they often go together, but the conclusions that were derived from either XANES or EXAFS were supporting different points of the logical sequence. Thus, XANES data were put in the beginning to reveal the oxidation state of the Tc, supporting the thought line, which opens the Results section, that reduction of Tc by ZVI had occurred. It is important to start with Tc reduction by iron, as this was the primary concept of the experiments – to reduce Tc with ZVI. On the other hand, EXAFS data answer the question of the possible Tc incorporation, and we first need to find out which iron mineral we should perform fit with, that is why results from other solid characterization techniques are better to be presented beforehand, to show a reader why we do fit with magnetite, not goethite or any other mineral. Thanks for this comment, and we also thought to present the XAFS analyses together, but we do consider making it rather an exception here and keep it apart for the better sequence of the data flow.

Comment 14

Upon inspection of the data, I fundamentally disagree with the conclusions drawn from this and I do believe these spectra show Tc incorporated into an iron oxide mineral phase. I have provided greater detail on some of the specifics of the data analysis below, however, a comparison with

previous published literature (Marshall et al., Environmental Science & Technology, 2014) clearly shows that the spectra provided in this study are not incorporated Tc species and suggests far more strongly that the phase present here is indeed TcO₂ with no Fe association. This can be seen clearly by the lack of large Fe peaks in the Fourier transform between 3 and 4 Å that are shown in Marshall et al. (2014) and that indicate Tc incorporation.

Response

Yes, we are familiar with the Marshall et al., Environmental Science & Technology, 2014 publication and we have referenced it as [15].

However, our data, unlike in the above and other publications, are based on several techniques, not only XAFS, the importance of such multimodal analysis to overcome limitations of a single method has been recently highlighted (Boglaenko, D. & Levitskaia, T. G. Abiotic Reductive Removal and Subsequent Incorporation of Tc(IV) into Iron Oxides: A Frontier Review. *Environ. Sci.: Nano* 6, 3492-3500 (2019)). We support the conclusions with the help of XPS and TEM, and, to the best of our knowledge, the combination of these techniques has not been previously performed for such system. Thus, the XPS spectra show three Tc species, one for Tc(VII) and two for Tc(IV), corresponding to two different local environments. It can be argued that the peaks we have assigned as a second environment for Tc(IV) could belong to another oxidation state of Tc, e.g. Tc(V) or Tc(VI) but these oxidation states are not supported by XANES analysis, and unlikely to be present in the absence of a stabilizing organic ligand. Furthermore, TEM analysis of the iron mineral area where Tc was observed with EDS, shows magnetite / maghemite crystalline structure (Fig. 6c) which, in case of co-precipitated TcO₂ would be distorted. The FIB vertical cross-section shows abundance of Tc in the vertical profile, not only on the surface of iron granules. We are inclined to defend our conclusion that Tc existed both as TcO₂ precipitate and structural Tc(IV).

We moved Figure “XPS Tc 3d spectra for granular ZVI contacted with 17 mM TcO₄⁻ in 80 mM NaCl for one month” to the main text (Figure 2) in support of the conclusion and removed it from the Supplementary information file.

We are giving here enlarged picture of the TEM of the Tc-abundant vertical cross-section of the granule (prepared by FIB). Light areas are concentrated with Tc areas, in which magnetite structure is clearly visible.

We consider TEM and XPS as a strong support of structural Tc existence. The need for the additional analytical methods was emphasized in the publication Boglaenko, D. & Levitskaia, T. G. Abiotic Reductive Removal and Subsequent Incorporation of Tc(IV) into Iron Oxides: A Frontier Review. *Environ. Sci.: Nano* 6, 3492-3500 (2019).

We kindly insist the reviewer re-evaluates this comment.

Comment 15

Additionally, why are the EXAFS (k-space) presented in k^2 weighting when the convention is k^3 ? This should be changed to make comparisons with other literature EXAFS data.

Response

We respectfully disagree with the reviewer because the information is the same and it does not affect fitting model and quality of fit. There are several publications where k^2 weighing was presented for the similar Tc studies: Heald et al. Radiochim. Acta (2012), Li et al. Chem Eng J (2019), Liu et al. Geochim Cosmochim Acta (2012), Zachara et al. Geochim Cosmochim Acta (2007).

Comment 16

Additionally, Table 2 (and Table 1 for that matter) doesn't need the reduced chi squared column as the number is meaningless unless comparing between fits of the same dataset (<https://bruceravel.github.io/demeter/documents/Athena/analysis/lcf.html> – the link provided is for LCF fitting but is true for EXAFS fitting too).

Response

References Marshall et al., Environmental Science & Technology (2014) and other (Lukens W. W. & Saslow S. A. *Dalton Trans.* (2018); Saslow, et al. *Environ. Sci. Technol.* (2017) in SI) provide reduced chi square and/or R-factor for different data sets, that is why we think it is important to be consistent with this convention.

Comment 17

The coordination numbers for the first Tc-O shell are much lower than the anticipated 6 fold coordination of Tc in a Tc(IV) oxidation state. Could the authors comment on why they believe this is the case?

Response

We bring the attention of the reviewer to the statement on the page 11: “However, our data did not result in a good fit with 6 atoms of O, possibly due to a fraction of Tc being on the surface of the iron oxide nanoparticles where the coordination numbers can be reduced.”

We repeated the fitting with fixed CN of O (=6), for which the quality of fit values are shown below in red, black are the original values reported in the manuscript:

Scan location	Chi _r ²	R-factor
ZVI-1-A	1032 (419)	0.03 (0.01)
ZVI-1-B	610 (33)	0.132 (0.008)
ZVI-2-A	389 (143)	0.05 (0.013)
ZVI-2-B	1445 113	0.09 0.006

Based on this analysis, we decided to keep the old fitting model.

Comment 18

The Debye-Waller factors for the first Fe shell (Tc-Fe1) in all samples, whilst not necessarily incorrect, seems to be quite large. Especially after consulting the literature (including the references provided in the manuscript) as other reported Debye-Waller factors are not this high. This could mean that this shell isn't really contributing to the fit and this shell should be F-tested to prove it is needed for a good fit. Whilst it may be convention to fix the Fe shells to a CN of 6, reducing these slightly may result in better fits and better fitting parameters. Further to this, if the samples in any way nanocrystalline or disordered, a decrease in CN of more distant shells would be expected.

Response

We re-evaluated the model and repeated fitting, but did not get better results. The Debye-Waller factors are indeed large, but consistent throughout the dataset, and we have to take into

consideration the mixed system, where the average number of Tc-O, Tc-Fe1, and Tc-Fe2 bonds represent the entire composition, where it can be, for example, half of the Tc bonds in the Fe₂O₃ (theoretical 6 Tc-Fe1 bonds) and half is in TcO₂ (zero Tc-Fe1 bonds). Thus, our number fixed to 6 bonds overestimates the real one and pushes sigma squared to the higher value. This is an inherent challenge of the heterogeneous system, where fixing the CNs to 6 will force a high value of sigma squared. We tried to hold the CNs to lower values, but it worsened the fit. We also tried to hold the sigma squared for the two paths equal and let the CNs vary but constrain them to be the same, and it did not help either.

Page 14, we clarified in the subsection EXAFS analysis of Tc K-edge in ZVI granules:

“Note that the large values of Tc-Fe1 σ^2 is an indication that the average CN of this path is lower, and is supportive of the heterogeneous system, where Tc coordinated not only with Fe in iron mineral, but with Tc in Tc⁴⁺ oxide.”

Discussion Comments

Comment 20

Is the pH stated (10.3) the final, stable pH? How did pH change throughout the experiment? Has the system reached equilibrium or could there still be transformations taking place?

Response

Data on pH, ORP and iron dissolution, together with Tc reduction kinetics are given in the preceding publication Boglaienko, et al. *J. Hazard. Mater.* **380**, 120836 (2019). High pH indicates on continuation of iron granules dissolution, according to reactions mentioned in the published work:

Unlike in the mentioned publication, where secondary precipitation (newly formed iron oxyhydroxides) prevailed and slightly decreased pH, stabilizing it around neutral, in this work, the initial iron concentration was 100 times higher, leading to much higher iron dissolution rates and increasing pH.

Comment 21

The discussion states ‘[the] the high dissolution rate of Fe²⁺ was expected to induce transformation of ferrihydrite into magnetite...’ but how do you know this when there is no aqueous geochemical data showing Fe dissolution?

Response

Data on iron dissolution are given in the preceding publication Boglaienko, et al. *J. Hazard. Mater.* **380**, 120836 (2019). In addition to this, we also conducted experiments (not published) with different amount of iron, and at larger concentrations, which were used here (50 g/L), the transformation to magnetite happened much faster, which is supported by the Cornell & Schwertmann (2003) book.

Page 4, information (with a reference [27]) on the previous studies for iron dissolution is given in the sentence:

“As it was investigated previously, ZVI materials manufacturing by different methods exhibit different kinetic of Tc⁷⁺ removal attributed to variable surface properties and rates of Fe⁰ oxidation and dissolution²⁷...”

Page 16, clarifying sentence was added in the Discussion section:

“Here, the initial amount of iron in the sample was high (50 g/L), leading to high Fe²⁺ dissolution rates that is supported by pH 10.3 indicative of Fe⁰ oxidation/dissolution reactions^{2,27}...”

Comment 22

Could the authors provide some clarity on how they determined the wt% Tc incorporated into magnetite as I am uncertain of how the value of 1.86 wt% is calculated. If, as the authors state, there is only 2-7 % magnetite in their system (according to PXRD), this would mean that of the 50 g/L ZVI added to the system, only 3.45 g/L are magnetite if take an approximate mean abundance of 5 % magnetite. Considering the authors state that approximately 32 % of the 17 mM Tc added to the experiment was incorporated this would mean 0.5 g/L Tc would need to be incorporated into 3.45 g/L of magnetite and the incorporation would be ~14 wt%. This value is much higher than previously reported values and likely indicates that the incorporated proportion of Tc into magnetite is incorrect.

Response

Your calculations are correct if to rely on PXRD data. However, PXRD cannot be taken as a precise quantitative estimate of magnetite, due to the high granularity of the sample precluding accurate quantification of iron phases. Mössbauer analysis gives more reliable information, pertaining to the whole sample used for the measurement. We respectfully ask the reviewer to reconsider these calculations.

Mossbauer analysis determined that sample contained around 42 % of magnetite. It yields around 28 g/L of magnetite and 0.5 g/L of Tc, which is ~1.9 wt%.

Page 16, in the main text it was noted in parenthesis:

“magnetite fraction was estimated *via* Mössbauer measurements of the one month reacted sample”.

Comment 23

The discussion needs more of clarity around the overall proposed pathways and mechanisms taking place here. Not just for the mineral transformations, but the pathways and fate of Tc. A concise, clear conclusions paragraph would achieve this.

Response

We believe that incorporated revisions significantly improved the clarity of discussion.

General Editorial Comments

Comment 24

Line 38 – ‘Further, formation of iron minerals is of singular complexity’ – I’m not overly clear what this statement means.

Response

Page 1: the sentence was clarified: “Further, formation of iron minerals is ~~of singular complexity complex~~ and, hence, ~~is of~~ fundamental and practical interest...”

Comment 25

Line 89 – ‘routs’ should be replaced with ‘routes’ if I’m inferring the statement correctly.

Response

Page 3: “routs” was replaced with “~~pathways~~”

Comment 26

Lines 291-292 – ‘junction’ should be replaced with ‘conjunction’ if I understand the sentence correctly.

Response

Page 12: “junction” was replaced with “~~conjunction~~”

Comment 27

Line 339 – ‘, Fig. 4,’ is inconsistent and should be changed to ‘(Fig. 4)’.

Response

Page 14: , Fig. 4, was changed to ~~(Fig. 4)~~

Reviewer 2

Comment 1

The risk of Tc99 contamination is lower not higher than that of Cs137 and Sr90. Only the long term extension of the risk for future generation is higher for Tc99.

Response

Page 2, the sentence was changed to:

“...(e.g. Chernobyl, Ukraine⁶), and **its environmental impact** is amplified by ⁹⁹Tc long half-life (213,000 years) and redox-dependent mobility with high solubility of the pertechnetate anion (TcO₄⁻) in aerobic conditions”.

Comment 2

This is not so new: Yalçintaş, E., Gaona, X., Scheinost, A.C., Kobayashi, T., Altmaier, M., Geckeis, H., 2015. Redox chemistry of Tc(VII)/Tc(IV) in dilute to concentrated NaCl and MgCl₂ solutions. *Radiochimica Acta* 103. <https://doi.org/10.1515/ract-2014-2272>,

Another is Kobayashi, T., Scheinost, A. C., Fellhauer, D., Gaona, X., Altmaier, D.: Redox behavior of Tc(VII)/Tc(IV) under various reducing conditions in 0.1MNaCl solutions. *Radiochim. Acta* 101, 323–332 (2013).

Response

We are familiar with the work mentioned above and the following have already been cited in the manuscript:

[16] Yalçintaş, E., Scheinost, A. C., Gaona, X. & Altmaier, M. Systematic XAS study on the reduction and uptake of Tc by magnetite and mackinawite. *Dalton Trans.* **45**, 17874–17885 (2016).

[30] Kobayashi 542 T., Scheinost A. C., Fellhauer D., Gaona X. & Altmaier M. Redox behaviour of Tc(VII)/Tc(IV) under various reducing conditions in 0.1 M NaCl solutions. *Radoichim Acta* **101**, 323–332 (2013).

We would like to emphasize that in contrast to the mentioned publications, in which the studied systems were anaerobic with relatively low Tc concentrations and controlled / buffered pH, our work was carried out in aerobic conditions with initial neutral pH allowed to vary and with much higher, almost 1000 times, Tc concentrations ($1.7 \cdot 10^{-2}$ M vs 10^{-5} M in the mentioned publications) leading to high iron loading with Tc.

Page 1, Abstract was clarified:

“Incorporation of Tc⁴⁺ into iron minerals has been studied predominantly for systems under conditions of carefully controlled **anaerobic** environment, employing syntheses with Fe²⁺ solutions or using pre-synthesized iron oxides/oxyhydroxides, and mechanisms of the

transformation of iron phases leading to incorporation of Tc⁴⁺ under aerobic conditions remain poorly understood.”

Comment 3

The concept of a waste form is not yet introduced in the text

Response

Page 3, Introduction section, the sentence was appended with clarification:

“ZVI (iron powder, nano-iron, and steel coupons) exhibits effective reductive separation of Tc^{25,26,27}, and is one of the materials compatible with immobilization and stabilization of the separated Tc for long-term disposal, but its application can be modulated by Tc anticorrosive properties^{28,29}, and has not been investigated for high Tc loading to achieve successful waste form design.”

Comment 4

It is not clear how NaCl promotes iron oxidation. NaCl does not contain oxidants and reducing Fe(II) is stable in NaCl solutions

Response

NaCl promotes iron oxidation via acceleration of electron transfer and densification of the electrical double layer (Cornell and Schwertmann 2003), and regeneration of reactive sites of iron (Yin et al. Chem Eng J 2012).

Comment 5

How do you know it is equilibrium?

Response

After about one month of reaction time we expect reduction-oxidation processes reaching steady state, however, we agree that it was not specifically studied, and we removed that from the text.

Page 3, Results section: the word “equilibrium” was removed from the sentence.

Comment 6

Give value

Response

Page 4, Results section: “0.08 mM” was added.

Comment 7

How do you know it is this phase?

Response

Based on the wealth of literature knowledge, hydrated TcO₂ oxide is the most likely product upon TcO₄⁻ reduction that can compose the precipitate. We agree that it was not analyzed and removed the statement from the text.

Comment 8

I do not really understand what one is seeing in fig b. What is the significance of the various colors? what is the link to the energy scale?

Response

These are the X-ray fluorescence maps that allow a reader to see where a beam scan was taken – two locations on one granule called ZVI-A, and two locations on another granule called ZVI-B. Colors reflect concentrations of Tc on the surface of each iron granule, and the scale on the right shows intensity of fluorescent x-ray with higher intensity assigned red color and number 1.0 and the lowest intensity dark blue color and number 0.001. Data presentation in a similar format was given in the publication by Lee et al. *Geochim. Cosmochim. Acta*, 136 (2014).

Page 5, for clarification, the Figure 1 caption was revised to include:

“XAFS scans were located on two ZVI particles, 1 and 2, in two locations, A and B. The color scales are arbitrary intensities of fluorescent X-rays with higher intensity assigned red color and number 1.0 and the lowest intensity dark blue color and number 0.001”.

Comment 9

This shows that the author are exposed to a highly dynamic system in which under oxidizing conditions Fe^{III} increases, but since there has remained some Fe⁰, Fe^{II} will also be present. The absence of clear Magnetite identification in presence of Tc makes the interpretation even more difficult

Response

We agree that it is a dynamic system, with Fe⁰ being continuously oxidized to Fe^{II} (given high initial Fe granules concentration of 50 g/L), however, the separate Fe^{II} phase is unstable in aerobic environment and its existence would be identified by Mossbauer both at ambient temperature and 77 K (doublet overlapping with the ferrihydrite peak around 0-3 mm/s) as it is seen in the publication Kukkadapu et al. *Geochim. Cosmochim. Acta*, 68 (2004). In our case, in all samples (short-term, long-term, with and without Tc) we did not observe a separate Fe^{II} phase, meaning that all Fe^{II} is going to be absorbed on ferrihydrite, inducing its transformation to magnetite, i.e. non-stoichiometric magnetite (partially oxidized in aerobic environment).

The text in page 16 is modified to add:

“Here, the initial amount of iron in the sample was high (50 g/L), leading to high Fe²⁺ dissolution rates that is supported by pH 10.3 indicative of Fe⁰ oxidation/dissolution reactions^{2,27}, which implies that transformation of ferrihydrite to magnetite would prevail over hematite or goethite. The PXRD and Mössbauer spectral evidence did not reveal separate Fe²⁺ mineral phase, e.g. siderite or green rust, that could have been expected due to presence of carbonate in aerated solution at elevated pH.”

Comment 10

The oxidizing effect of Tc(VII) need to be described as well

Response

In our study Tc(VII) exhibited anticorrosive effects, slowing down transformation of ferrihydrite to magnetite, which is described in Discussion (page 16) and supported by citations 28 and 29.

Comment 11

Hence it is impossible to distinguish between these states?

Response

Yes, such differentiation is very difficult, that is why studies by other techniques complementary to EXAFS are recommended to support the conclusions. Here, we performed XPS and TEM measurements supporting presence of incorporated Tc(IV) phase.

Comment 12

This is not really an incorporation in magnetite but an inclusion of one phase in another (no coprecipitate) such behavior depends strongly on the precipitation conditions and cannot be generalized. I do not believe that this is just mono and dimers as these would not be identified as clusters by STEM or EDS

Response

We have considered this possibility, but based on the results of three techniques, EXAFS, XPS, and TEM, we are inclined to conclude incorporated phase, as there are two environments for Tc(IV) in XPS fit, and EXAFS fit was the most successful upon inclusion of octahedral Tc-Fe paths. Fit with the TcO₂ model, that would be predominant in case of the phase inclusion, did not produce good results.

Comment 13

It is aerobic, but the Fe is a source for reduction. This is a very unstable system and evolves depending on the quantity of air circulation, the specific surface area of initial ZVI, the mixing etc. The slower transformation to magnetite in presence of Tc is not really discussed mechanistically, one may invoke an oxidizing effect

Response

The Fe oxidation and phase transformation in the systems with and without Tc was studied under identical conditions allowing their comparison, and the obtained results are consistent with the previously reported anticorrosive properties of pertechnetate anion (see references 28 and 29 in the manuscript). Presence of anions changes kinetics of iron mineral transformation (Cornell and Schwertmann 2003), however the oxidizing effect of TcO₄⁻ and its effect on iron mineral formation has not been studied before. Here, in this work, we focused mainly on the Tc(VII) reduction and solid characterization of iron mineral phases formed in presence and absence of high loading of Tc. We agree with the comment that further studies are warranted to elucidate the mechanisms of the TcO₄⁻ effect on iron oxidation and transformation, which are outside the scope of the current investigation.

Comment 14

the impact of pH excursion is not discussed

Response

Observed high pH (10.3) implies that transformation path of ferrihydrite to magnetite would go faster, as it does at pH > 9 (Cornell and Schwertmann 2003) (please see the second paragraph of the Discussion section). Moreover, at this pH the presence of carbonate in the aerated solution can lead to formation of siderite (FeCO₃), but it was not observed by any solid characterization analyses, i.e. characteristic doublet would be present for the Fe(II) phase in the Mossbauer spectra (which was not found).

Several reactions of iron dissolution and iron mineral phases transformation affect the pH, and we inclined to think that iron dissolution at our high initial Fe(0) concentrations would prevail (Boglaienko, et al. *J. Hazard. Mater.* 380 (2019)):

Page 16, Discussion section, sentences were added:

“Here, the initial amount of iron in the sample was high (50 g/L), leading to high Fe²⁺ dissolution rates that is supported by pH 10.3 indicative of Fe⁰ oxidation/dissolution reactions^{2,27}, which implies that transformation of ferrihydrite to magnetite would prevail over hematite or goethite. The PXRD and Mössbauer spectral evidence did not reveal separate Fe²⁺ mineral phase, e.g. siderite or green rust, that could have been expected due to presence of carbonate in aerated solution at elevated pH.”

Comment 15

What is this?

Response

“Thermodynamically spontaneous system” term is used here to say that none of the thermodynamic parameters are controlled, and system is allowed to transform without application of the external stimuli or constrains.

Comment 16

Not at all understandable what is meant by these cloudy works: if spontaneous means "uncontrolled", I would strongly argue against waste form development and remediation without control

Response

The subject of this investigation is of fundamental importance and does not involve development of a particular waste form or remediation method.

Comment 17

No dimer has been identified in the present study

Response

We emphasize heterogeneity of the incorporated phase of Tc, and compare it to the other reported studies, including Tc(IV) dimers. We did not state that we identified dimers.

Reviewer 3

This paper describes the phenomena associated with the interaction of pertechnetate with metal iron and the change of oxidation states that follow. It is a well-written document reporting the work of a highly diverse team of scientists on multiple techniques.

The major claim of the paper is the quantitative reduction of Tc(VII) by iron and oxidation of iron to magnetite.

This paper provide a substantial advance in science and in the field of nuclear waste management.

Overall, this is a high-quality paper that has its place in Communications Chemistry.

I have a few comments:

Comment 1

Line 99-100: The authors point out that the chloride ion will not influence the system because of Cl⁻ inability to complex. While chloride complexation with cation is minimal, it is not entirely inexistent. Moreover, the designation of “Tc7+” is misleading, as heptavalent Tc is in the form of the anionic species TcO₄⁻ and does not act as a cation. Finally, speaking of counter anion of concern, the authors should discuss the influence of the presence of NH₄⁺, present in the system at the same concentration as Tc, at 17 mM.

Response

Tc is changed to “**TcO₄⁻**”.

Ammonia affects iron solubility via complex formation only in highly concentrated solutions, 4.5 M NH₄OH and higher (Klocke and Hixson. Solubility of ferrous iron in aqueous ammoniacal solutions. Ind. Eng. Chem. Process Des. Dev. 1972, 11, 1, 141-146). In our solution we have no greater than 0.017 M NH₄⁺, we do not expect significant complexing effect from ammonium cation. Moreover, at our pH (10.3), measured about a month since the start of the experiment, NH₄⁺ would have partitioned to a gas phase as NH₃ (pK_a 9.24).

Comment 2

Figure 2, line 242: I am surprised the subsequent magnetic shows so much crystallinity. Under such mild condition of pressure, one should expect an amorphous solid. I would like to see the author providing some insight on the matter.

Response

Reaction time of one month provides enough time for oxidized iron phases transformation and formation of crystalline phases.

Comment 3

Table 1: I find the errors provided uncomfortably low.

Response

Table 1 was appended with LCF for EXAFS spectra, and the errors are comparable (the same order of magnitude). Unlike the reduced chi squared for EXFAS fitting, it is much lower in the linear combination analysis.

Reviewers' comments:

Reviewer #1 (Remarks to the Author):

Firstly, I would like to thank the author's for the considerable amount of effort and time they've put into comprehensively revising the manuscript. I feel that the changes and additions to the text have helped clarify a number of issues I raised in the paper and I feel like I can follow and understand the work much more clearly. Additionally, the comments that the authors have provided in response to my concerns are also greatly appreciated and are well detailed. That being said, I do still have a couple of points that may want to be considered before moving forward with any publication of this work (see attached document for comments).

Reviewer #2 (Remarks to the Author):

The authors claim that they show a new method for reducing the environmental impact of Tc in the environment in remediation situations or waste form fabrication. However, while unrealistically high Tc(VII) concentrations can be reduced by 99.8 %, the remaining concentrations of 1E-5 M/L are still highly toxic (see my comment in the attached paper). Despite my remarks in the last review, the paper still talks about equilibrium conditions or concentrations, while this is definitely a non equilibrium system which, as it is under oxic conditions will continue to release Tc(VII) by reoxidation. In consequence and despite the nice work of sample characterizations, and in particular related to the none-proven claims, I think the paper cannot be published

Reviewer #3 (Remarks to the Author):

the authors have resolved my comments.

Response to Rebuttal

General Comments

Firstly, I would like to thank the author's for the considerable amount of effort and time they've put into comprehensively revising the manuscript. I feel that the changes and additions to the text have helped clarify a number of issues I raised in the paper and I feel like I can follow and understand the work much more clearly. Additionally, the comments that the authors have provided in response to my concerns are also greatly appreciated and are well detailed. That being said, I do still have a couple of points that may want to be considered before moving forward with any publication of this work.

Comments 7-9

Thank you for providing this additional analysis. Indeed the LCF fitting of the EXAFS is very consistent with the XANES which is a good source of further evidence for your conclusions. For clarity, I think it may be useful to add the PCA to the SI and LCF table provided in response to **Comment 7** as I think it provides useful information to support conclusions drawn from both the XANES and EXAFS.

Comments 14, 17 and 18

I believe the addition of the LCF of EXAFS and the PCA has helped to abate some of my concerns of the EXAFS fitting. Following the clarification and additions to the manuscript, I think I now understand the reasons for the fits produced much better and that actually having 3 different oxidation states (including 2 Tc(IV)). This being said, I still have some significant reservations about the EXAFS fitting.

With respect to the first Tc-O shell, I appreciate that the system is both heterogeneous and nanoparticulate, however, the reduction in CN for this shell is, in one fit, greater than 50 % of the expected value (2.7). I have experience dealing with nanocrystalline materials in EXAFS myself and whilst I agree that a reduction in CN is possible, this fitting suggests that over 50 % of the material does not contain a Tc-O bond length. This seems contradictory to the fact that both Fe shells are then fixed at a CN 6, as (and I appreciate it's a mixed phase system and fitting is complex) you would expect a similar decrease in CN for these shells.

If we consider the LCF and PCA to inform the EXAFS fits, we should expect ~10-20 % Tc(VII) (as the pertechnetate ion), ~50 % Tc(IV)O₂, and the remaining ~30-40 % as Tc(IV) in magnetite. Firstly, there is no accountancy for Tc(VII) in the fits as one would expect some contribution at ~1.73 Å. Whilst this is only a small portion of the system, it's not unreasonable to suggest that there would be 0.4-0.8 CN for this shell (and this may help with the overall fit, in particular with the first Tc-O shell). As the remaining 80-90 % of the system is Tc(IV), one would expect the Tc-O shell to be ~5. A small reduction due to nanoparticulate or surface bound nature would probably bring the CN down to around 4 but I'm unsure it would produce CN of ~3. These rough calculations can be extended to the Fe shells (which I understand have been fixed with some justification, however, I'm unsure if this is correct) and one would expect CN of ~2-2.5. Additionally, as the peaks in the FT for Fe are a very small contribution (as mentioned in my previous comments), these could possibly be being produced by multiple scattering pathways rather than having a full contribution from Fe. This has been studied previously by Sherman *et al.*¹ on a system containing U on goethite (I appreciate this is a different element but the principles are worth considering to inform the fit).

I realise the authors have justified their reasoning for the fits provided now (which is helpful and I thank them for this), however, I do believe that sometimes going for a slightly worse

quantitative/qualitative fit in favour of a more thorough and robust fit in terms of chemistry is more valid.

To summarise, overall I think there is now more evidence for the 3 phase fitting model, however I'm still unsure about the validity of the EXAFS fits as there seems to be some issues (highlighted) and possible evidence of over-fitting the data. Whilst there is some support from XPS/TEM for the conclusions drawn, I'm still not sure that the EXAFS models provided are correct.

References

[1] Sherman, D. M. et al., Surface complexation of U(VI) on goethite (α -FeOOH), *Geochimica et Cosmochimica Acta*, **2008**, 72(2), 298-310.

Response to Reviewers' comments:

TITLE: Spontaneous Tc⁷⁺ – Fe⁰ Redox Continuum: Sequestered Tc⁴⁺ Clusters and Retarded Mineral Transformation of Iron
Communications Chemistry

Reviewer 1:

General Comments

Firstly, I would like to thank the author's for the considerable amount of effort and time they've put into comprehensively revising the manuscript. I feel that the changes and additions to the text have helped clarify a number of issues I raised in the paper and I feel like I can follow and understand the work much more clearly. Additionally, the comments that the authors have provided in response to my concerns are also greatly appreciated and are well detailed. That being said, I do still have a couple of points that may want to be considered before moving forward with any publication of this work.

Response

We are tankful for the kind comments and careful review.

Comment 1

Thank you for providing this additional analysis. Indeed the LCF fitting of the EXAFS is very consistent with the XANES which is a good source of further evidence for your conclusions. For clarity, I think it may be useful to add the PCA to the SI and LCF table provided in response to Comment 7 as I think it provides useful information to support conclusions drawn from both the XANES and EXAFS.

Response:

We added PCA figures to SI, page 3, Supplementary Figure 2, and made a reference to it in the main text:

Page 4, a sentence was amended:

“Moreover, principal component analysis (PCA) was supportive of three components (Supplementary Fig. 2), hence, three standards were kept for LCF...”

We also added the LCF table to SI, page 14, Supplementary Table 1, and mentioned it in the main text:

Page 4:

“... LCF combinatorics showed better statistical results for three components, with R-factor and Chi,² almost twice smaller for three components vs two components (Supplementary Table 1).”

Comment 2

I believe the addition of the LCF of EXAFS and the PCA has helped to abate some of my concerns of the EXAFS fitting. Following the clarification and additions to the manuscript, I think I now understand the reasons for the fits produced much better and that actually having 3 different oxidation states (including 2 Tc(IV)). This being said, I still have some significant reservations about the EXAFS fitting. With respect to the first Tc-O shell, I appreciate that the system is both heterogeneous and nanoparticulate, however, the reduction in CN for this shell is, in one fit, greater than 50 % of the expected value (2.7). I have experience dealing with nanocrystalline materials in EXAFS myself and whilst I agree that a reduction in CN is possible, this fitting suggests that over 50 % of the material does not contain a Tc-O bond length. This seems contradictory to the fact that both Fe shells are then fixed at a CN 6, as (and I appreciate it's a mixed phase system and fitting is complex) you would expect a similar decrease in CN for these shells. If we consider the LCF and PCA to inform the EXAFS fits, we should expect ~10-20 % Tc(VII) (as the pertechnetate ion), ~50 % Tc(IV)O₂, and the remaining ~30-40 % as Tc(IV) in magnetite. Firstly, there is no accountancy for Tc(VII) in the fits as one would expect some contribution at ~1.73 Å. Whilst this is only a small portion of the system, it's not unreasonable to suggest that there would be 0.4-0.8 CN for this shell (and this may help with the overall fit, in particular with the first Tc-O shell). As the remaining 80-90 % of the system is Tc(IV), one would expect the Tc-O shell to be ~5. A small reduction due to nanoparticulate or surface bound nature would probably bring the CN down to around 4 but I'm unsure it would produce CN of ~3. These rough calculations can be extended to the Fe shells (which I understand have been fixed with some justification, however, I'm unsure if this is correct) and one would expect CN of ~2-2.5. Additionally, as the peaks in the FT for Fe are a very small contribution (as mentioned in my previous comments), these could possibly be being produced by multiple scattering pathways rather than having a full contribution from Fe. This has been studied previously by Sherman et al.¹ on a system containing U on goethite (I appreciate this is a different element but the principles are worth considering to inform the fit). I realise the authors have justified their reasoning for the fits provided now (which is helpful and I thank them for this), however, I do believe that sometimes going for a slightly worse quantitative/qualitative fit in favour of a more thorough and robust fit in terms of chemistry is more valid. To summarise, overall I think there is now more evidence for the 3 phase fitting model, however I'm still unsure about the validity of the EXAFS fits as there seems to be some issues (highlighted) and possible evidence of over-fitting the data. Whilst there is some support from XPS/TEM for the conclusions drawn, I'm still not sure that the EXAFS models provided are correct.

References

[1] Sherman, D. M. et al., Surface complexation of U(VI) on goethite (α -FeOOH), *Geochimica et Cosmochimica Acta*, 2008, 72(2), 298-310.

Response:

Coordination numbers for Tc-O are indeed low, however, in samples ZVI-1-A and ZVI-2-A they are 4.4 and 4.1 respectively, which agrees with the reviewer's analysis in the comment. Sample ZVI-1-B has CN = 3.1, and it is the only sample ZVI-2-B, which has CN = 2.7, but we think it is important to be consistent with the modelling for all our samples. Moreover, for these two

samples, scan locations were at regions of higher abundance of Tc (false-image fluorescence images in Fig. 1b), and LCF showed almost twice higher amount of TcO_4^- , which was not in the fit for this path (Tc-O distance is around 1.7 Å).

Page 13, a sentence was added:

“While CN for ZVI-1-A and ZVI-2-A (4.4 and 4.1 respectively) are reasonable, accounting for the mixture of Tc^{7+} and Tc^{4+} and surface fraction of Tc^{4+} , CN for ZVI-1-B and ZVI-2-B are quite low (3.1 and 2.7 respectively), but the data of these scan locations represent elevated amount of Tc (false-color images with Tc abundance on ZVI granules, Fig. 1b), with higher fraction of TcO_4^- , *i.e.* 0.21 and 0.20 respectively (Table 1), contribution from which is not included in the fit.”

Reviewer 2

Comment 1

I still maintain that thermodynamic equilibrium has not been attained and cannot be attained in an aerobic open system.

Response

Page 2, the sentence was edited and thermodynamic equilibrium was removed:

“... in a system ~~that is allowed thermodynamically equilibrate~~ under ambient conditions where only limited number of controlled parameters are imposed to resemble natural pathways.”

Comment 2

I maintain: this sentence suggests that the study focusses on equilibrium. However, the study has shown no equilibrium of an iron system at equilibrium: under oxic conditions iron or magnetite cannot be at equilibrium!!!

Response

Page 3, the sentence was edited to eliminate the equilibria statement:

“Thus, the behavior of an iron system ~~at equilibria with the environment~~ requires additional investigation for each unique setting and contaminant of concern.”

Comment 3

This is not an equilibrium Tc concentrations since equilibrium was not reached. 0.01 mM is still a high concentration from public health perspective: it is 1000 times higher than the solubility limited concentration with respect to Tc(IV) under reducing conditions. This solution has 6.2E5 Bq/L and a person which drinks this over a year will get a dose by ingestion of about 0.4 Sv/yr. Now you can say that actual concentrations in the environment are probably lower, so the dose will also be lower, but the paper provides no data as to whether at lower concentration the same apparent efficiency of the process can be maintained.

Response

Page 3, the sentence was edited:

“~~Concentration of Tc~~ in the supernatant was 0.01 mM ~~indicating nearly quantitative removal~~ of TcO_4^- .”

Reviewers' comments:

Reviewer #1 (Remarks to the Author):

I'd like to thank the author's for taking into account my first comment from my previous review. However, I feel the response to my comment concerning the EXAFS fitting wasn't adequately addressed and has unfortunately raised more questions.

I agree that the CN for Tc-O in ZVI-1-A and ZVI-2-A are acceptable, however, the CN for ZVI-1-B and ZVI-2-B are not and I do not agree with the author's methodology for implementing the fitting parameters here.

I understand that there is a need to have a relatively consistent fitting model throughout a system that is being studied (a global fit if you will), however, if the 'global' fit doesn't work for all the samples it means that either the fit is incorrect, and/or the samples that are being studied are significantly different. Either way, this means that the samples need to be fitted with a different fitting model (or at least tested with another to then prove the validity of the original model). After reading the authors' response to my comment I'm now more concerned by the absence of the contribution at 1.7 Å for TcO₄⁻. As the authors highlight in the response, there is even a higher fraction of TcO₄⁻ in these samples (~20 %) so it's fundamentally incorrect to discount the contribution of this shell from the EXAFS fit (the LCF performed by the authors of the EXAFS even shows that there is a necessary 10-15 % contribution from TcO₄⁻ in order to produce a good fit). Additionally, the authors also point out that these samples are at a point of significantly higher Tc concentration and so this provides more evidence suggesting that a different fit should be applied to the 'A' samples.

Apart from wanting to maintain a global fit for all samples, what is the chemical reasoning behind omitting a scattering shell from the EXAFS fitting model that is clearly present as shown by LCF of both the XANES and the EXAFS? Even if the addition of this shell doesn't improve the fit, it's good practise to add this shell, F-test its contribution, and provide some SI evidence to discount this.

Additionally, I don't feel that my comments regarding multiple scattering pathways as alternatives to Fe shells was addressed and I believe this to be an important consideration when producing valid EXAFS models for these systems.

Reviewer #2 (Remarks to the Author):

It is OK now

Response to Reviewers' comments:

TITLE: Spontaneous $\text{Tc}^{7+} - \text{Fe}^0$ Redox Continuum: Sequestered Tc^{4+} Clusters and Retarded Mineral Transformation of Iron

Communications Chemistry

Reviewer #1 (Remarks to the Author):

I'd like to thank the author's for taking into account my first comment from my previous review. However, I feel the response to my comment concerning the EXAFS fitting wasn't adequately addressed and has unfortunately raised more questions.

Comment 1

I agree that the CN for Tc-O in ZVI-1-A and ZVI-2-A are acceptable, however, the CN for ZVI-1-B and ZVI-2-B are not and I do not agree with the author's methodology for implementing the fitting parameters here.

I understand that there is a need to have a relatively consistent fitting model throughout a system that is being studied (a global fit if you will), however, if the 'global' fit doesn't work for all the samples it means that either the fit is incorrect, and/or the samples that are being studied are significantly different. Either way, this means that the samples need to be fitted with a different fitting model (or at least tested with another to then prove the validity of the original model). After reading the authors' response to my comment I'm now more concerned by the absence of the contribution at 1.7 Å for TcO_4^- . As the authors highlight in the response, there is even a higher fraction of TcO_4^- in these samples (~20 %) so it's fundamentally incorrect to discount the contribution of this shell from the EXAFS fit (the LCF performed by the authors of the EXAFS even shows that there is a necessary 10-15 % contribution from TcO_4^- in order to produce a good fit). Additionally, the authors also point out that these samples are at a point of significantly higher Tc concentration and so this provides more evidence suggesting that a different fit should be applied to the 'A' samples.

Apart from wanting to maintain a global fit for all samples, what is the chemical reasoning behind omitting a scattering shell from the EXAFS fitting model that is clearly present as shown by LCF of both the XANES and the EXAFS? Even if the addition of this shell doesn't improve the fit, it's good practise to add this shell, F-test its contribution, and provide some SI evidence to discount this.

Response

We attempted to remodel the data for the most disputed sample, more precisely, different scan spot ZVI-2-B. In the table below please find the original fit, then versions 2 and 3 (v.2 and v.3), which had a path for Tc-O that comes from pertechnetate, in addition to the original fit (only the

new path is reported in the Table below, but the other paths were kept as before). As the pertechnetate contributes around 15-20%, we might expect CN around 0.5-1.0 and interatomic distance approximately 1.7 Å. As you can see, interatomic distances are off significantly (2.23 and -3.22 Å) with large errors (in parenthesis on the last two significant figures). Then we fixed Tc-O CN value (previous 2.7) to 4 and Tc-O2 CN to 1 (v.4). Interatomic distances are better, however the error for σ^2 is still large; statistically this fit is much worse, giving χ_r^2 of 1032 (vs 113 of the original fit). Finally, we discarded the path from the pertechnetate and fixed CN for Tc-O to 4 (v.5). The fit is acceptable, except χ_r^2 of 510 (vs 113 of the original fit).

We consider our original fit to be reasonable and consistent with the overall modeling approach. Please, take into consideration what ZVI-1-A, ZVI-1-B and ZVI-2-A, ZVI-2-B represent. As it shown in Fig. 1b (X-ray fluorescence maps in the manuscript), ZVI-1 is a granule of iron, and ZVI-2 is another granule of iron; A and B are different locations within the same granule, those are not different samples. Additionally, the rest of parameters (atomic distance R, Debye-Waller factor, and shift in energy) for these two data sets, which are questioned by the reviewer, agree very well with the model for the rest of the data sets.

Scan location	Path	CN	R (Å)	σ^2 (Å ²)	ΔE_0 (eV)	χ_r^2	R-factor
ZVI-2-B original	Tc-O	2.7	1.99 (1)	0.001 (1)	-1.10 ± 1.16	113	0.006
	Tc-Fe1	6.0 ^f	3.09 (1)	0.021 (3)			
	Tc-Fe2	6.0 ^f	3.48 (2)	0.013 (3)			
	Tc-Tc	0.8	2.64 (1)	0.003 (2)			
Fit v.2	Tc-O2	0.5	2.23 (22)	0.007 (49)		520	0.005
Fit v.3	Tc-O2	1.0	-3.22 (30)	0.072 (66)		397	0.004
Fit v.4	Tc-O	4.0	1.98 (3)	0.005 (3)		1032	0.010
	Tc-O2	1.0	1.61 (17)	0.010 (22)			
Fit v.5	Tc-O	4.0 ^f	1.98 (2)	0.005 (2)	$-2.98 \pm (2.59)$	510	0.027
	Tc-Fe1	6.0 ^f	3.06 (1)	0.022 (3)			
	Tc-Fe2	6.0 ^f	3.45 (4)	0.015 (3)			
	Tc-Tc	0.8	2.62 (1)	0.002 (4)			

The authors of the current draft of manuscript published a review paper, which examined similar systems of the incorporated vs precipitated Tc(IV) and which stated that EXAFS evidence should not be solely relied upon and should be complemented with another techniques (Boglaenko and Levitskaia, Environ Sci: Nano, 2019). Unlike the other studies, heavily relying on EXAFS data analysis mainly, including the study referred to by the reviewer in round 1 (Marshall et al. Environ. Sci. Technol., 2014), we designed a comprehensive study, where other techniques

bring significant evidence of Tc(IV) incorporated and Tc(IV) precipitated. The evidence of the Tc(IV) incorporation into iron oxide crystal structure at such high loadings that were not studied before is supported by XPS and TEM, as well as supported by the visual evidence from EDS and SEM techniques. Adjustment of the number (CN) of oxygens surrounding Tc atom in the EXAFS fit model, argued upon, does not change the course of the conclusions. The CNs are not given as the exact estimates, but with an error (mentioned in the Table 2 footnote of the manuscript), and it is known that unless CN is fixed, it can vary (Ravel, B. and Kelly, S.D., 2007, February. The difficult chore of measuring coordination by EXAFS. In *AIP Conference Proceedings* (Vol. 882, No. 1, pp. 150-152). American Institute of Physics). EXAFS data analysis here brings not the only, but the additional or supplemental support to the existence of the incorporated Tc(IV) fraction. The model was made to reflect that fraction of the incorporated Tc(IV) into magnetite crystal structure, and it has good statistical quality (reduced Chi-square and R-factor).

Comment 2

Additionally, I don't feel that my comments regarding multiple scattering pathways as alternatives to Fe shells was addressed and I believe this to be an important consideration when producing valid EXAFS models for these systems.

Response

We considered including multiple scattering, double scattering paths from O – O (R_{eff} 3.44) in TcO₂ and double scattering path from O – Tc (R_{eff} 3.17). We chose these paths as their R_{eff} are close to those modelled with Fe shell in magnetite and can serve as alternatives. However, for O – O path σ^2 was 2.638 and ΔR was -5.72, which are not acceptable values; and for O – Tc path σ^2 was -0.005, which is also not acceptable. Statistically this fit was also much worse, $\text{Chi}_r^2 = 2091$ vs 113 of the original fit. These values are not reportable, as σ^2 cannot be negative and ΔR should be below 0.1.

Scan location	Path	CN	R (Å)	σ^2 (Å ²)	ΔE_0 (eV)	Chi_r^2	R-factor
ZVI-2-B	Tc-O	6	1.98	0.012	-3.1	2091	0.12
With	Tc-Tc	1	2.59	0.004			
TcO2	Tc-O-Tc	4	3.25	-0.005			
	Tc-O-O	4	-2.28	2.638			

Plot in K-space with contribution from each path is given below:

New fit with multiple scattering paths from TcO₂:

Old fit that was initially in the manuscript:

“merge diamond hot” is the data; that was the name of the scan spot given by our former colleague, who collected data in Advanced Photon Source, as the granule ZVI-2 reminded her diamond shape, and hot referred to high concentration of Tc radioisotope.

We show that TcO_2 model does not fit our data, proving the presence of incorporated Tc(IV) into magnetite lattice, which is also confirmed by the other techniques given in the manuscript.

Page 14, a sentence was added:

“Additionally, a model with contribution from the multiple scattering paths as an alternative to the iron shell scattering was tested and produced non-reportable results, supporting the evidence of incorporated Tc^{4+} .”

REVIEWERS' COMMENTS:

Reviewer #1 (Remarks to the Author):

I'd like to thank the author's for taking the time and effort to thoroughly test the alternative fitting models and indeed, the fits provided in the manuscript were the best possible fits, in spite of some unusual nuances to the models. The excellent rebuttal document has addressed my concerns regarding the fits. Again, I'd like to thank the author's for all their hard work throughout this process.